# Physics-Embedded Neural Networks: Graph Neural PDE Solvers with Mixed Boundary Conditions

**Masanobu Horie**
RICOS Co. Ltd.
University of Tsukuba
horie@ricos.co.jp

**Naoto Mitsume**
University of Tsukuba
mitsume@kz.tsukuba.ac.jp

## Abstract

Graph neural network (GNN) is a promising approach to learning and predicting physical phenomena described in boundary value problems, such as partial differential equations (PDEs) with boundary conditions. However, existing models inadequately treat boundary conditions essential for the reliable prediction of such problems. In addition, because of the locally connected nature of GNNs, it is difficult to accurately predict the state after a long time, where interaction between vertices tends to be global. We present our approach termed physics-embedded neural networks that considers boundary conditions and predicts the state after a long time using an implicit method. It is built based on an $E(n)$-equivariant GNN, resulting in high generalization performance on various shapes. We demonstrate that our model learns flow phenomena in complex shapes and outperforms a well-optimized classical solver and a state-of-the-art machine learning model in speed-accuracy trade-off. Therefore, our model can be a useful standard for realizing reliable, fast, and accurate GNN-based PDE solvers. The code is available at https://github.com/yellowshippo/penn-neurips2022.

## 1  Introduction

Partial differential equations (PDEs) are of interest to many scientists because of their application in various fields such as mathematics, physics, and engineering. Numerical analysis is used to solve PDEs because most PDE problems in real life cannot be solved analytically. For example, predicting fluid behavior in complex shapes is an essential topic because it is helpful for product design, disaster reduction, weather forecasting, and many others; however, it is a difficult problem and takes time to solve using classical solvers. Machine learning is a promising approach to predicting such phenomena because it can utilize data similar to the state to be predicted, while classical solvers cannot.

However, the main challenge in dealing with complex phenomena such as fluids is to guarantee generalization performance because possible states in complex systems can be huge and may not be covered using a purely data-driven approach. Therefore, we must apply appropriate inductive biases to machine learning models. Many approaches successfully introduced various inductive biases such as local connectedness using graph neural networks (GNNs) and symmetry under coordinate transformations using equivariance.

While these methods have made great progress in solving PDEs using machine learning, there is still room for improvement. First, there is need for an efficient and provable way to respect boundary conditions like Dirichlet and Neumann, i.e., mixed boundary conditions. Rigorous fulfillment of Dirichlet boundary conditions is indispensable because they are hard constraints and different Dirichlet conditions correspond to different problems users would like to solve. Second, there is need to reinforce the treatment of global interaction to predict the state after a long time, where interactions

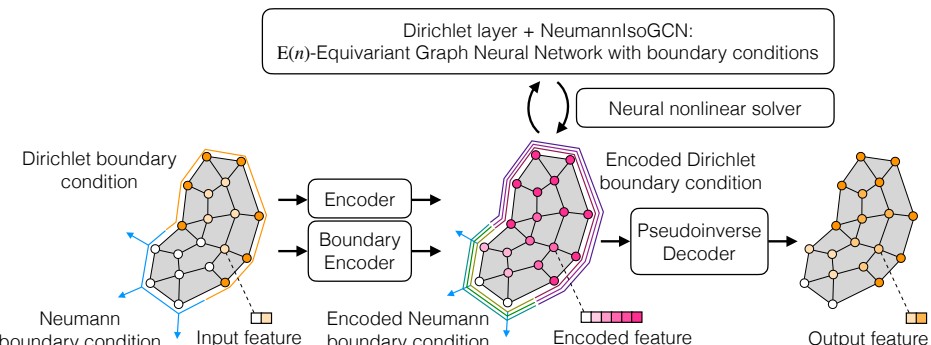

Figure 1: Overview of the proposed method. On decoding input features, we apply boundary encoders to boundary conditions. Thereafter, we apply a nonlinear solver consisting of an E($n$)-equivariant graph neural network in the encoded space. Here, we apply encoded boundary conditions for each iteration of the nonlinear solver. After the solver stops, we apply the pseudoinverse decoder to satisfy Dirichlet boundary conditions.

tend to be global. GNNs have excellent generalization properties because of their locally-connected nature; however, they may miss global interaction due to their localness.

We propose physics-embedded neural networks (PENNs), a machine learning framework to address these issues by embedding physics in the models. We build our model based on IsoGCN (Horie et al., 2021), a lightweight E($n$)-equivariant GNN to reflect physical symmetry and realize fast prediction. Furthermore, we construct a method to consider mixed boundary conditions. Finally, we reconsider a way to stack GNNs based on a nonlinear solver, which naturally introduces the global pooling to GNNs as the global interaction with high interpretability. In experiments, we demonstrate that our treatment of Neumann boundary conditions improves the predictive performance of the model, and our method can fulfill Dirichlet boundary conditions with no error. Our method also achieves state-of-the-art performance compared to a classical, well-optimized numerical solver and a baseline machine learning model in speed-accuracy trade-off. Figure 1 shows the overview of the proposed model. Our main contributions are summarized as follows:

- We construct models to satisfy mixed boundary conditions: the *boundary encoder*, *Dirichlet layer*, *pseudoinverse decoder*, and *NeumannIsoGCN* (NIsoGCN). The considered models show provable fulfillment of boundary conditions, while existing models cannot.

- We propose *neural nonlinear solvers*, which realize global connections to stably predict the state after a long time.

- We demonstrate that the proposed model shows state-of-the-art performance in speed-accuracy trade-off, and all the proposed components are compatible with E($n$)-equivariance.

## 2   Background and related work

In this section, we review the foundation of PDEs to clarify the problems we solve and introduce related works where machine learning models are used to solve PDEs.

### 2.1   Partial differential equations (PDEs) with boundary conditions

A general form of the $d$-dimensional temporal PDEs that we consider can be expressed as follows:

$$\frac{\partial \boldsymbol{u}}{\partial t}(t, \boldsymbol{x}) = \mathcal{D}(\boldsymbol{u})(t, \boldsymbol{x}) \qquad (t, \boldsymbol{x}) \in (0, T) \times \Omega, \qquad (1)$$

$$\boldsymbol{u}(t = 0, \boldsymbol{x}) = \hat{\boldsymbol{u}}_0(\boldsymbol{x}) \qquad \boldsymbol{x} \in \Omega, \qquad (2)$$

$$\boldsymbol{u}(t, \boldsymbol{x}) = \hat{\boldsymbol{u}}(t, \boldsymbol{x}) \qquad (t, \boldsymbol{x}) \in (0, T) \times \partial\Omega_{\text{Dirichlet}}, \qquad (3)$$

$$\hat{\boldsymbol{f}}(\nabla \boldsymbol{u}(t, \boldsymbol{x}), \boldsymbol{n}(\boldsymbol{x})) = \boldsymbol{0} \qquad (t, \boldsymbol{x}) \in (0, T) \times \partial\Omega_{\text{Neumann}}, \qquad (4)$$

where $\Omega$ is the domain, $\partial\Omega$ is the boundary of $\Omega$, and $\partial\Omega_{\text{Dirichlet}}$ and $\partial\Omega_{\text{Neumann}}$ are boundaries with Dirichlet and Neumann (mixed) boundary conditions. $\hat{\cdot}$ is a known function, and $\mathcal{D}$ is a known nonlinear differential operator, which can be nonlinear and contain spatial differential operators (see equation 18 for an example of $\mathcal{D}$). $\boldsymbol{n}(\boldsymbol{x})$ is the normal vector at $\boldsymbol{x} \in \partial\Omega$. Equation 3 is called the Dirichlet boundary condition, where the value on $\partial\Omega_{\text{Dirichlet}}$ is set as a constraint. Equation 4 corresponds to the Neumann boundary condition, where the value of the derivative $\boldsymbol{u}$ in the direction of $\boldsymbol{n}$ is set on $\partial\Omega_{\text{Neumann}}$ rather than the value of $\boldsymbol{u}$. When $\boldsymbol{u} : (0, T) \times \Omega \to \mathbb{R}^f$ satisfies Equations 1 – 4, it is called the solution of the (initial-) boundary value problem.

### 2.1.1 Discretization

PDEs are defined in a continuous space to make differentials meaningful. Discretization can be applied in space and time so that computers can solve PDEs easily. In numerical analysis of complex-shaped domains, we commonly use meshes (discretized data of shapes), which can be regarded as graphs. We denote the position of the $i$th vertex as $\boldsymbol{x}_i$ and the value of a function $f$ at the $\boldsymbol{x}_i$ as $f_i$.[1]

The simplest method to discretize time is the explicit Euler method formulated as:
$$\boldsymbol{u}(t + \Delta t, \boldsymbol{x}_i) = \boldsymbol{u}(t, \boldsymbol{x}_i) + \mathcal{D}(\boldsymbol{u})(t, \boldsymbol{x}_i)\Delta t, \tag{5}$$
which updates $\boldsymbol{u}(t, \boldsymbol{x}_i)$ with a small increment $\mathcal{D}(\boldsymbol{u})(t, \boldsymbol{x}_i)\Delta t$. Another way to have time discretization is the implicit Euler method formulated as:
$$\boldsymbol{u}(t + \Delta t, \boldsymbol{x}_i) = \boldsymbol{u}(t, \boldsymbol{x}_i) + \mathcal{D}(\boldsymbol{u})(t + \Delta t, \boldsymbol{x}_i)\Delta t, \tag{6}$$
which solves Equation 6 rather than simply updating variables to ensure the original PDE is satisfied numerically. The equation can be viewed as a nonlinear optimization problem by formulating it as:
$$\boldsymbol{R}(\boldsymbol{v}) := \boldsymbol{v} - \boldsymbol{u}(t, \cdot) - \mathcal{D}(\boldsymbol{v})\Delta t, \tag{7}$$
$$\text{Solve}_{\boldsymbol{v}} \ \boldsymbol{R}(\boldsymbol{v})(\boldsymbol{x}_i) = \boldsymbol{0}, \ \forall i, \tag{8}$$
where $\boldsymbol{R}(\boldsymbol{v})$ is the residual vector of the discretized PDE. The solution of Equation 8 corresponds to $\boldsymbol{u}(t + \Delta t, \boldsymbol{x})$. By letting $\nabla\phi = \boldsymbol{R}$ for an appropriate $\phi$, solving Equation 8 corresponds to optimizing $\phi$ in an $(f \times n)$-dimensional space, where $n$ is the number of vertices in the considered mesh. A simple way to solve such an optimization problem is to apply gradient descent formulated as:
$$\boldsymbol{v}^{(0)} := \boldsymbol{u}(t, \cdot), \quad \boldsymbol{v}^{(i+1)} := \boldsymbol{v}^{(i)} - \alpha^{(i)}\boldsymbol{R}(\boldsymbol{v}^{(i)}), \tag{9}$$
where $\alpha^{(i)} \in \mathbb{R}$ is determined using line search. However, due to the high computational cost of the search, $\alpha$ can be fixed to a small value, which corresponds to the explicit Euler method with the time step size $\alpha\Delta t$. Barzilai & Borwein (1988) suggested another simple yet effective way to determine the step size using a two-point approximation to the secant equation underlying quasi-Newton methods.

## 2.2 Neural PDE solvers

We review machine learning models used to solve PDEs called neural PDE solvers, typically formulated as $\boldsymbol{u}(t_{n+1}, \boldsymbol{x}_i) \approx \mathcal{F}_{\text{NN}}(\boldsymbol{u})(t_n, \boldsymbol{x}_i)$ for $(t_n, \boldsymbol{x}_i) \in \{t_0, t_1, \dots\} \times \Omega$, where $\mathcal{F}_{\text{NN}}$ is a machine learning model.

### 2.2.1 Physics-informed neural networks (PINNs)

Raissi et al. (2019) made a pioneering work combining PDE information and neural networks, called PINNs, by adding loss to monitor how much the output satisfies the equations. PINNs can be used to solve forward and inverse problems and extract physical states from measurements (Pang et al., 2019; Mao et al., 2020; Cai et al., 2021). However, PINNs' outputs should be functions of space because PINNs rely on automatic differentiation to obtain loss regarding PDEs. This design constraint significantly limits the model's generalization ability because the solution of a PDE could be entirely different when the shape of the domain or boundary condition changes. Besides, the loss reflecting PDEs helps models learn physics at training time; however, prediction by PINN models can be out of physics because of lacking PDE information inside the model. Therefore, these methods are not applicable in building models that are generalizable over shape and boundary condition variations. As seen in Section 3, our model contains PDE information inside and does not take absolute positions of vertices, thus resulting in high generalizability (See Figure 3).

---

[1] Strictly speaking, components of the PDE e.g. $\mathcal{D}$ and $\Omega$ can be different before and after discretization. However, we use the same notation regardless of discretization to keep the notation simple.

### 2.2.2 Graph neural network based PDE solvers

As discussed in Section 2.1.1, one can regard a mesh as a graph. GNNs can take any graphs as inputs (Gori et al., 2005; Scarselli et al., 2008; Kipf & Welling, 2017; Gilmer et al., 2017), having the possibility to generalize over various graphs, i.e., meshes. Therefore, GNNs are strong candidates for learning mesh-structured numerical analysis data, as seen in Alet et al. (2019); Chang & Cheng (2020); Pfaff et al. (2021). Brandstetter et al. (2022) advanced these works for efficient and stable prediction. Their method could also consider boundary conditions by feeding them to the models as inputs. Here, one could expect the model to learn to satisfy boundary conditions approximately, while there is no guarantee to fulfill hard constraints such as Dirichlet conditions. In contrast, our model ensures the satisfaction of boundary conditions. Besides, most GNNs use local connections with a fixed number of message passings, which lacks consideration of global interaction. We suggest an effective way to incorporate a global connection with GNN through the neural nonlinear solver.

### 2.2.3 Equivariant models

In addition to GNNs, another essential concept to help machine learning models generalize is equivariance. Equivariance is characterized by using group action as $f(g \cdot x) = g \cdot f(x)$ for $f : X \to Y$ and $g \in G$ acting on $X$ and $Y$. In particular, $E(n)$-equivariance is essential to predict the solutions of physical PDEs because it describes rigid body motion, i.e., translation, rotation, and reflection. Ling et al. (2016) and Wang et al. (2021) introduced equivariance to a simple neural network and CNN to predict flow phenomena. Both works showed that equivariance improved predictive and generalization performance compared to models without equivariance. Horie et al. (2021) proposed $E(n)$-equivariant GNNs based on GCNs (Kipf & Welling, 2017), called IsoGCNs. A form of their model is formulated as:

$$[\nabla \psi]_i \approx [\text{IsoGCN}_{0 \to 1}(\psi)]_i := \left[ \sum_{l \in \mathcal{N}_i} \frac{\boldsymbol{x}_l - \boldsymbol{x}_i}{\|\boldsymbol{x}_l - \boldsymbol{x}_i\|} \otimes \frac{\boldsymbol{x}_l - \boldsymbol{x}_i}{\|\boldsymbol{x}_l - \boldsymbol{x}_i\|} \right]^{-1} \sum_{j \in \mathcal{N}_i} \frac{\psi_j - \psi_i}{\|\boldsymbol{x}_j - \boldsymbol{x}_i\|} \frac{\boldsymbol{x}_j - \boldsymbol{x}_i}{\|\boldsymbol{x}_j - \boldsymbol{x}_i\|} \boldsymbol{W},$$

(10)

where $\mathcal{N}_i$ is the neighborhood of the $i$th vertex, $\otimes$ is the tensor product operator, and $\boldsymbol{W}$ is a trainable matrix acting on feature index. Here, we denote $\text{IsoGCN}_{0 \to 1}$ an IsoGCN layer that converts the input scalar (rank-0 tensor) field $\psi$ to the output vector (rank-1 tensor) field. This layer corresponds to the gradient operator, which helps learn PDEs because spatial derivatives such as gradient play an essential role in PDEs. They applied the model to the heat equation problem, showing high predictive performance and fast prediction, while boundary condition treatment was out of their scope.

## 3 Proposed method

We present our model architecture. We adopt an encode-process-decode architecture, proposed by Battaglia et al. (2018), which has been applied successfully in various previous works, e.g., Horie et al. (2021); Brandstetter et al. (2022). Our key concept is to encode input features, including information on boundary conditions, apply a GNN-based nonlinear solver loop reflecting boundary conditions in the encoded space, then decode carefully to satisfy boundary conditions in the output space.

### 3.1 Dirichlet boundary model

As demonstrated theoretically and experimentally in literature (Hornik, 1991; Cybenko, 1992; Nakkiran et al., 2021), the expressive power of neural networks comes from encoding in a higher-dimensional space, where the corresponding boundary conditions are not trivial. However, if there are no boundary condition treatments in layers inside the processor, which resides in the encoded space, the trajectory of the solution can be far from the one with boundary conditions. Therefore, boundary condition treatments in an encoded space are essential for obtaining reliable neural PDE solvers that fulfill boundary conditions.

To ensure the same encoded space between variables and boundary conditions, we use the same encoder for variables and the corresponding Dirichlet boundary conditions, which we term the *boundary encoder*, as follows:

$$\boldsymbol{h}_i = \boldsymbol{f}_{\text{encode}}(\boldsymbol{u}_i) \text{ in } \Omega, \quad \hat{\boldsymbol{h}}_i = \boldsymbol{f}_{\text{encode}}(\hat{\boldsymbol{u}}_i) \text{ on } \partial\Omega_{\text{Dirichlet}}$$

(11)

One can easily apply Dirichlet boundary conditions in the aforementioned encoded space using the *Dirichlet layer* defined as:

$$\text{DirichletLayer}(\boldsymbol{h}_i) = \begin{cases} \boldsymbol{h}_i, & \boldsymbol{x}_i \notin \partial\Omega_{\text{Dirichlet}} \\ \hat{\boldsymbol{h}}_i, & \boldsymbol{x}_i \in \partial\Omega_{\text{Dirichlet}} \end{cases} \tag{12}$$

This process is necessary to return to the state respecting the boundary conditions after some operations in the processor, which might disrespect the conditions.

After the processor layers, we decode the hidden features using functions satisfying:

$$\boldsymbol{f}_{\text{decode}} \circ \boldsymbol{f}_{\text{encode}}(\hat{\boldsymbol{u}}_i) = \hat{\boldsymbol{u}}_i \text{ on } \partial\Omega_{\text{Dirichlet}} \tag{13}$$

This condition ensures that the encoded boundary conditions correspond to the ones in the original physical space. Demanding that Equation 13 holds for arbitrary $\hat{\boldsymbol{u}}$; we obtain $\boldsymbol{f}_{\text{decode}} \circ \boldsymbol{f}_{\text{encode}} = \text{Id}_{\boldsymbol{u}}$, resulting in $\boldsymbol{f}_{\text{decode}} = \boldsymbol{f}_{\text{encode}}^+$, which we call the *pseudoinverse decoder*. It is pseudoinverse because $\boldsymbol{f}_{\text{encode}}$, in particular encoding in a higher-dimensional space, may not be invertible. Therefore, we construct $\boldsymbol{f}_{\text{encode}}^+$ using pseudoinverse matrices. For more details, see Appendix A.1.

## 3.2 Neumann boundary model

Matsunaga et al. (2020) proposed a wall boundary model to deal with Neumann boundary conditions for the least squares moving particle semi-implicit (LSMPS) method (Tamai & Koshizuka, 2014), a framework to solve PDEs using particles. The LSMPS method is the origin of the IsoGCN's gradient operator, so one can imagine that the wall boundary model may introduce a sophisticated treatment of Neumann boundary conditions into IsoGCN. We modified the wall boundary model to adapt to the situation where the vertices are on the Neumann boundary, which differs from the situation of particle simulations (see Appendix A.2 for more details). Our formulation of IsoGCN with Neumann boundary conditions, which is termed *NeumannIsoGCN* (NIsoGCN), is expressed as:

$$\text{NIsoGCN}_{0\to1}(\psi) := \boldsymbol{M}_i^{-1}\left[\sum_{j\in\mathcal{N}_i}\frac{\psi_j - \psi_i}{\|\boldsymbol{x}_j - \boldsymbol{x}_i\|}\frac{\boldsymbol{x}_j - \boldsymbol{x}_i}{\|\boldsymbol{x}_j - \boldsymbol{x}_i\|} + w_i\boldsymbol{n}_i\hat{g}_i\right]\boldsymbol{W} \tag{14}$$

$$\boldsymbol{M}_i := \sum_{l\in\mathcal{N}_i}\frac{\boldsymbol{x}_l - \boldsymbol{x}_i}{\|\boldsymbol{x}_l - \boldsymbol{x}_i\|}\otimes\frac{\boldsymbol{x}_l - \boldsymbol{x}_i}{\|\boldsymbol{x}_l - \boldsymbol{x}_i\|} + w_i\boldsymbol{n}_i\otimes\boldsymbol{n}_i, \tag{15}$$

where $\hat{g}_i$ is the value of the Neumann boundary condition at $\boldsymbol{x}_i$, $\boldsymbol{W}$ is a trainable matrix, and $w_i > 0$ is an untrainable parameter to control the strength of the Neumann constraint. As $w_i \to \infty$, the model strictly satisfies the given Neumann condition in the direction $\boldsymbol{n}_i$, while the directional derivatives in the direction of $(\boldsymbol{x}_j - \boldsymbol{x}_i)$ tend to be relatively neglected. Thus, we keep the value of $w_i$ moderate to consider derivatives in both $\boldsymbol{n}$ and $\boldsymbol{x}$ directions. In particular, we set $w_i = 10.0$, assuming that around ten vertices may virtually exist "outside" the boundary on a flat surface in a 3D space.

NIsoGCN is a straightforward generalization of the original IsoGCN by letting $\boldsymbol{n}_i = \boldsymbol{0}$ when $\boldsymbol{x}_i \notin \partial\Omega_{\text{Neumann}}$. This model can also be generalized to vectors or higher rank tensors, similarly to the original IsoGCN's construction (see Appendix A.2). Therefore, NIsoGCN can express any spatial differential operator, constituting $\mathcal{D}$ in PDEs.

## 3.3 Neural nonlinear solver

As reviewed in Section 2.1, one can regard solving PDEs as optimization. Here, we adopt the Barzilai–Borwein method (Barzilai & Borwein, 1988) to solve Equation 8 in the encoded space. In our case, the step size $\alpha^{(i)}$ of gradient descent is approximated as:

$$\alpha^{(i)} \approx \alpha_{\text{BB}}^{(i)} := \frac{\langle\boldsymbol{h}^{(i)} - \boldsymbol{h}^{(i-1)}, \boldsymbol{R}(\boldsymbol{h}^{(i)}) - \boldsymbol{R}(\boldsymbol{h}^{(i-1)})\rangle_\Omega}{\langle\boldsymbol{R}(\boldsymbol{h}^{(i)}) - \boldsymbol{R}(\boldsymbol{h}^{(i-1)}), \boldsymbol{R}(\boldsymbol{h}^{(i)}) - \boldsymbol{R}(\boldsymbol{h}^{(i-1)})\rangle_\Omega}, \tag{16}$$

wherer $\boldsymbol{R}(\boldsymbol{h})$ is the residual vector in the encoded space and $\langle\boldsymbol{f},\boldsymbol{g}\rangle_\Omega := \sum_{\boldsymbol{x}_i\in\Omega}\boldsymbol{f}(\boldsymbol{x}_i)\cdot\boldsymbol{g}(\boldsymbol{x}_i)$ denotes the inner product over the mesh. Because the inner product is taken all over the mesh (graph), computing $\alpha_{\text{BB}}^{(i)}$ corresponds to global pooling. With that view, one can find similarities between Equation 9 and deep sets (Zaheer et al., 2017), which is a successful method to learn point cloud data and has a strong background regarding permutation equivariance. For more details, see Appendix A.3.

Table 1: MSE loss ($\pm$ the standard error of the mean) on test dataset of gradient prediction. $\hat{g}_{\text{Neumann}}$ is the loss computed only on the boundary where the Neuman condition is set.

| Method | $\nabla\phi(\times 10^{-3})$ | $\hat{g}_{\text{Neumann}}(\times 10^{-3})$ |
|---|---|---|
| Original IsoGCN | $192.72 \pm 1.69$ | $1390.95 \pm 7.93$ |
| **NIsoGCN** (Ours) | $6.70 \pm 0.15$ | $3.52 \pm 0.02$ |

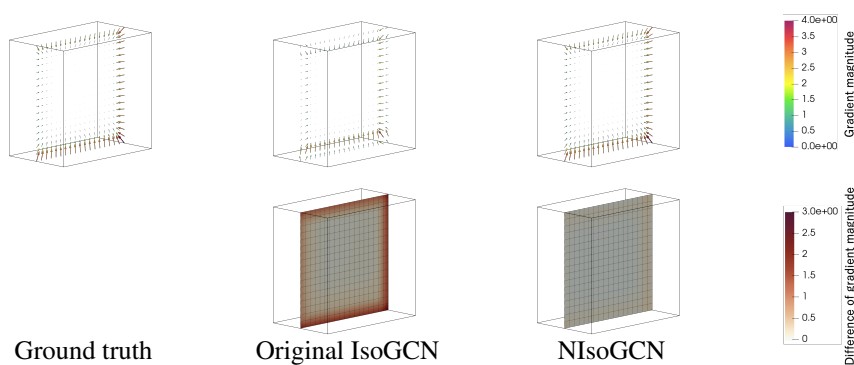

Figure 2: Gradient field (top) and the magnitude of error between the predicted gradient and the ground truth (bottom) of a test data sample, sliced on the center of the mesh.

Our aim is to use Equation 16, approximating the nonlinear differential operator $\mathcal{D}$ in Equation 7 with NIsoGCN. By doing this, we expect the processor to consider both local and global information, which may have an advantage over simply stacking GNNs corresponding to the explicit method as discussed in Section 2.1.1. Combinations of solvers and neural networks are already suggested in, e.g., NeuralODE (Chen et al., 2018). The novelty of our study is the extension of existing methods for solving PDEs with spatial structure and the incorporation of global pooling into the solver, enabling us to capture global interaction, which we refer to as the *neural nonlinear solver*. Finally, the update from the state at the $i$th iteration $\boldsymbol{h}^{(i)}$ to the $(i+1)$th in the neural nonlinear solver is expressed as:

$$\boldsymbol{h}^{(i+1)} = \text{DirichletLayer}\left(\boldsymbol{h}^{(i)} - \alpha_{\text{BB}}^{(i)}\left[\boldsymbol{h}^{(i)} - \boldsymbol{h}^{(0)} - \mathcal{D}_{\text{NIsoGCN}}(\boldsymbol{h}^{(i)})\Delta t\right]\right), \qquad (17)$$

where $\boldsymbol{h}^{(0)}$ is the encoded $\boldsymbol{u}(t,\cdot)$ reflecting Equation 9 and $\mathcal{D}_{\text{NIsoGCN}}$ is an E($n$)-equivariant GNN reflecting the structure of $\mathcal{D}$ using differential operators provided by NIsoGCN. Here, Equation 17 enforces hidden features to satisfy the encoded PDE, including boundary conditions, motivating us to call our model *physics-embedded neural networks* because it embeds physics (PDEs) in the model rather than in the loss.

## 4 Experiments

Using numerical experiments, we demonstrate the proposed model's validity, expressibility, and computational efficiency. We use two types of datasets: 1) the gradient dataset to verify the correctness of NIsoGCN and 2) the incompressible flow dataset to demonstrate the speed and accuracy of the model. We also present ablation study results to corroborate the effectiveness of the proposed method. The implementation of our model is based on the original IsoGCN's code.[2] Our implementation is available online.[3] All the details of the experiments and another simple experiment can be found in Appendix B, C, and D.

### 4.1 Gradient dataset

As done in Horie et al. (2021), we conducted experiments to predict the gradient field from a given scalar field to verify the expressive power of NIsoGCN. We generated cuboid-shaped meshes

---

[2] https://github.com/yellowshippo/isogcn-iclr2021, Apache License 2.0.

[3] https://github.com/yellowshippo/penn-neurips2022, Apache License 2.0.

Table 2: MSE loss (± the standard error of the mean) on test dataset of incompressible flow. If "Trans." is "Yes," it means evaluation is done on randomly rotated and transformed test dataset. $\hat{}_{\text{Dirichlet}}$ is the loss computed only on the boundary where the Dirichlet condition is set for each $\boldsymbol{u}$ and $p$. MP-PDE's results are based on the time window size equaling 40 as it showed the best performance in the tested MP-PDEs. For complete results, see Table 4.

| Method | Trans. | $\boldsymbol{u}$ $(\times 10^{-4})$ | $p$ $(\times 10^{-3})$ | $\hat{\boldsymbol{u}}_{\text{Dirichlet}}$ $(\times 10^{-4})$ | $\hat{p}_{\text{Dirichlet}}$ $(\times 10^{-3})$ |
|---|---|---|---|---|---|
| MP-PDE TW = 20 | No | **1.30** ± 0.01 | 1.32 ± 0.01 | 0.45 ± 0.01 | 0.28 ± 0.02 |
| | Yes | 1953.62 ± 7.62 | 281.86 ± 0.78 | 924.73 ± 6.14 | 202.97 ± 3.81 |
| **PENN** (Ours) | No | 4.36 ± 0.03 | **1.17** ± 0.01 | **0.00** ± 0.00 | **0.00** ± 0.00 |
| | Yes | **4.36** ± 0.03 | **1.17** ± 0.01 | **0.00** ± 0.00 | **0.00** ± 0.00 |

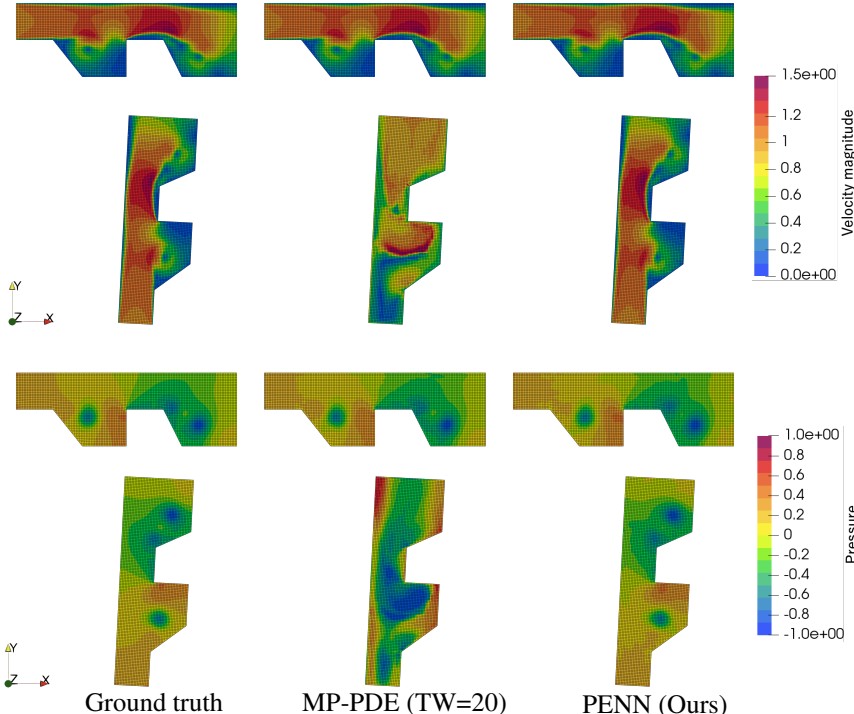

Ground truth      MP-PDE (TW=20)      PENN (Ours)

Figure 3: Comparison of the velocity field (top two rows) and the pressure field (bottom two rows) without (first and third rows) and with (second and fourth rows) random rotation and translation. PENN prediction is consistent under rotation and translation due to the E($n$)-equivariance nature of the model, while MP-PDE's predictive performance degrades under transformations.

randomly with 10 to 20 cells in the X, Y, and Z directions. We then generated random scalar fields over these meshes using polynomials of degree 10 and computed their gradient fields analytically. Our training, validation, and test datasets consisted of 100 samples. Table 1 and Figure 2 show that the proposed NIsoGCN improves gradient prediction, especially near the boundary, showing that our model successfully considers Neumann boundary conditions.

## 4.2 Incompressible flow dataset

We tested the expressive power our model by learning incompressible flow in complex shapes. The corresponding nonlinear differential operator is denoted as:

$$\mathcal{D}_{\text{NS}}(\boldsymbol{u}) := -(\boldsymbol{u} \cdot \nabla)\boldsymbol{u} + \frac{1}{\text{Re}} \nabla \cdot \nabla \boldsymbol{u} - \nabla p, \tag{18}$$

with the incompressible condition $\nabla \cdot \boldsymbol{u} = 0$, where, in the present case, $\boldsymbol{u}$ is the flow velocity field, $p$ is the pressure field, and $\mathrm{Re}$ is the Reynolds number.

### 4.2.1 Data

To generate the dataset, we first generated pseudo-2D shapes, with one cell in the Z direction, by changing design parameters, starting from three template shapes. Thereafter, we performed numerical analysis using a classical solver, OpenFOAM,[4] with $\Delta t = 10^{-3}$, and the initial conditions were the solutions of potential flow, which can be computed quickly and stably using the classical solver. The Reynolds number $\mathrm{Re}$ was around $10^3$. The linear solvers used were generalized geometric-algebraic multi-grid for $p$ and the smooth solver with the Gauss–Siedel smoother for $\boldsymbol{u}$. Template shapes, design parameters, and boundary conditions used can be found in Appendix C.2.

To confirm the expressive power of the proposed model, we used coarse input meshes for machine learning models. We generated these coarse meshes by setting cell sizes roughly four times larger than the original numerical analysis. We obtained ground truth variables using interpolation. The task was to predict flow velocity and pressure fields at $t = 4.0$ using information available before numerical analysis, e.g., initial conditions and the geometries of the meshes. Training, validation, and test datasets consisted of 203, 25, and 25 samples, respectively. We generated the dataset by randomly rotating and translating test samples to monitor the generalization ability of machine learning models.

### 4.2.2 Machine learning models

We constructed the PENN model corresponding to the incompressible Navier–Stokes equation. In particular, we adopted the fractional step method, where the pressure field was also obtained as a PDE solution along with the velocity field. We encoded each feature in a 4, 8, or 16-dimensional space. After features were encoded, we applied a neural nonlinear solver containing NeumanIsoGCNs and Dirichlet layers, reflecting the fractional step method (See Equations 51 and 52). Inside the nonlinear solver's loop, we had a subloop that solved the Poisson equation for pressure, which also reflected the considered PDE (See Equation 50). We looped the solver for pressure five times and four or eight times for velocity. After these loops stopped, we decoded the hidden features to obtain predictions for velocity and pressure, using the corresponding pseudoinverse decoders.

For the state-of-the-art baseline model, we selected MP-PDE (Brandstetter et al., 2022) as it also provides a way to deal with boundary conditions. We used the authors' code[5] with minimum modification to adapt to the task. We tested various time window sizes such as 2, 4, 10, and 20, where one step corresponds to time step size $\Delta t = 0.1$. With changes in time window size, we changed the number of hops considered in one operation of the GNN of the baseline to have almost the same number of hops visible from the model when predicting the state at $t = 4.0$. The numbers of hidden features, 32, 64, and 128, were tested. All models were trained for up to 24 hours using one GPU (NVIDIA A100 for NVLink 40GiB HBM2).

### 4.2.3 Results

Table 2 and Figure 3 show the comparison between MP-PDE and PENN. The predictive performances of both models are at almost the same level when evaluated on the original test dataset. The results show the great expressive power of the MP-PDE model because we kept most settings at default as much as possible and applied no task-specific tuning. However, when evaluating them on the transformed dataset, the predictive performance of MP-PDE significantly degrades. Nevertheless, PENN shows the same loss value up to the numerical error, confirming our proposed components are compatible with $\mathrm{E}(n)$-equivariance. In addition, PENN exhibits no error on the Dirichlet boundaries, showing that our treatment of Dirichlet boundary conditions is rigorous.

Figure 4 shows the speed-accuracy trade-off for OpenFOAM, MP-PDE, and PENN. We varied mesh cell size, the time step size, linear sover settings for OpenFOAM to have different computation speeds and accuracy. The proposed model achieved the best performance in speed-accuracy trade-off between all the tested methods under fair comparison conditions.

---

[4] https://www.openfoam.com/
[5] https://github.com/brandstetter-johannes/MP-Neural-PDE-Solvers

Table 3: Ablation study on the incompressible flow dataset. The value represents MSE loss ($\pm$ standard error of the mean) on the test dataset. "Divergent" means the implicit solver does not converge and the loss gets extreme value ($\sim 10^{14}$).

| Method | $\boldsymbol{u}$ ($\times 10^{-4}$) | $p$ ($\times 10^{-3}$) | $\hat{\boldsymbol{u}}_{\text{Dirichlet}}$ ($\times 10^{-4}$) | $\hat{p}_{\text{Dirichlet}}$ ($\times 10^{-3}$) |
|---|---|---|---|---|
| Without encoded boundary | Divergent | Divergent | Divergent | Divergent |
| Without boundary condition in the neural nonlinear solver | $65.10 \pm 0.38$ | $21.70 \pm 0.09$ | $0.00 \pm 0.00$ | $0.00 \pm 0.00$ |
| Without neural nonlinear solver | $31.03 \pm 0.19$ | $9.81 \pm 0.04$ | $\mathbf{0.00} \pm 0.00$ | $\mathbf{0.00} \pm 0.00$ |
| Without boundary condition input | $20.08 \pm 0.21$ | $3.61 \pm 0.02$ | $59.60 \pm 0.89$ | $1.43 \pm 0.05$ |
| Without Dirichlet layer | $8.22 \pm 0.07$ | $1.41 \pm 0.01$ | $18.20 \pm 0.28$ | $0.38 \pm 0.01$ |
| Without pseudoinverse decoder | $8.91 \pm 0.06$ | $2.36 \pm 0.02$ | $1.97 \pm 0.06$ | $\mathbf{0.00} \pm 0.00$ |
| Without pseudoinverse decoder with Dirichlet layer after decoding | $6.65 \pm 0.05$ | $1.71 \pm 0.01$ | $\mathbf{0.00} \pm 0.00$ | $\mathbf{0.00} \pm 0.00$ |
| **PENN** | $\mathbf{4.36} \pm 0.03$ | $\mathbf{1.17} \pm 0.01$ | $\mathbf{0.00} \pm 0.00$ | $\mathbf{0.00} \pm 0.00$ |

Table 3 presents the results of the ablation study. Comparison between models with and without the proposed components shows that the proposed components, i.e., the boundary encoder, Dirichlet layer, pseudoinverse decoder, and neural nonlinear solver, significantly improve the models. The neural nonlinear solver in the encoded space turned out to have the biggest impact on the performance, while the Dirichlet layer ensured reliable models that strictly respect Dirichlet boundary conditions.

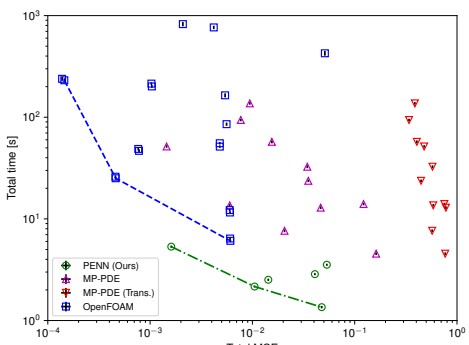

Figure 4: Comparison of computation time and total MSE loss ($\boldsymbol{u}$ and $p$) on the test dataset (with and without transformation) between OpenFOAM, MP-PDE, and PENN. The error bar represents the standard error of the mean. All computation was done using one core of Intel Xeon CPU E5-2695 v2@2.40GHz. Data used to plot this figure are shown in Tables 5, 6, and 7.

## 5   Conclusion

We have presented an $\text{E}(n)$-equivariant, GNN-based neural PDE solver, PENN, which can fulfill boundary conditions required for reliable predictions. The model has superiority in embedding the information of PDEs (physics) in the model and speed-accuracy trade-off. Therefore, our model can be a useful standard for realizing reliable, fast, and accurate GNN-based PDE solvers. Although the property of our model is preferable, it also limits the applicable domain of the model because we need to be familiar with the concrete form of the PDE of interest to construct the effective PENN model. For instance, the proposed model cannot exploit its potential to solve inverse problems where explicit forms of the governing PDE are not available for such tasks. Therefore, combining PINNs and PENNs could be the next direction of the research community.

## 6   Potential negative societal impacts

We have built a foundation to learn PDEs in a steerable manner rather than focusing on a specific application. Because of that, we envisage minimal risk of direct abusing our present work. However, as mentioned in the introduction, solving PDEs has many impacts on various domains, from both positive and negative aspects. Thus, our work and possible successive ones may be abused, aiming to harm lives and the environment. Therefore, the research community, including us, must be careful in using them and control the research direction to prevent abusing these technologies.

## Acknowledgments and Disclosure of Funding

This work was supported by PRESTO, Japan Science and Technology Agency, Grant Number JPMJPR21O9, Japan and by the New Energy and Industrial Technology Development Organization, Grant Number JPNP14012, Japan.

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
