# A  Details of the proposed method

## A.1  Construction of pseudoinverse decoder

We can construct the pseudoinverse decoders for a wide range of neural network architectures. For instance, the pseudoinverse decoder for an multilayer perceptron (MLP) with one hidden layer $f(x) = \sigma_2\left(W_2\sigma_1(W_1 x + b_1) + b_2\right)$ can be constructed as:

$$f^+(h) = W_1^+ \sigma_1^{-1}\left(W_2^+ \sigma_2^{-1}(h) - b_2\right) - b_1, \tag{19}$$

where $W^+$ is the pseudoinverse matrix of $W$ and $\sigma$ is an invertible activation function whose $\mathrm{Dom}(\sigma) = \mathrm{Im}(\sigma) = \mathbb{R}$. We chose LeakyReLU

$$\mathrm{LeakyReLU}(x) = \left\{ \begin{array}{ll} x & (x \geq 0) \\ ax & (x < 0), \end{array} \right. \tag{20}$$

where set $a = 0.5$ because an extreme value of $a$ (e.g., 0.01) could lead to an extreme value of gradient for the inverse function. In addition, one may choose activation functions whose $\mathrm{Im}(\sigma) \neq \mathbb{R}$, such as $\tanh$. However, in that case, we must ensure that the input value to the pseudoinverse decoder is in $\mathrm{Im}(\sigma)$ (in case of $\tanh$, it is $(-1, 1)$); otherwise, the computation would be invalid.

Besides, similar to the Dirichlet encoder and pseudoinverse decoder, we could define the specific encoder and decoder for the Neumann boundary condition. However, this is not included in the contributions of our work because it does not improve the performance of our model, which may be because the Neumann boundary condition is a soft constraint in contrast to the Dirichlet one and expressive power seems more important than that inductive bias.

## A.2  Derivation of NIsoGCN

Matsunaga et al. (2020) derived a gradient model that can treat the Neumann boundary condition with an arbitrary convergence rate with regard to spatial resolution. Here, we derive our gradient model, i.e., NIsoGCN, in a different way to simplify the discussion because we only need the first-order approximation for fast computation.

Before deriving NIsoGCN, we review introductory linear algebra using simple normation. Using a unit basis $\{e_i \in \mathbb{R}^d : \|e_i\| = 1\}_{i=1}^d$, one can decompose a vector $v \in \mathbb{R}^d$ using:

$$v = \sum_i (v \cdot e_i) e_i. \tag{21}$$

Now, consider replacing the basis $\{e_i \in \mathbb{R}^d\}_{i=1}^d$ with a set of vectors $B = \{b_i \in \mathbb{R}^d\}_{i=1}^{d'}$, called a *frame*, that spans the space but is not necessarily independent (thus, $d' \geq d$). Using the frame, one can assume $v$ is decomposed as:

$$v = \sum_i (v \cdot b_i) A b_i, \tag{22}$$

where $A$ is a matrix that corrects the "overcount" that may occur using the frame (for instance, consider expanding $(1, 0)^\top$ with the frame $\{(1,0)^\top, (-1,0)^\top, (0,1)^\top\}$). A set $\{A b_i\}_{i=0}^{d'}$ is called a *dual frame* for $B$. We can find the concrete form of $A$ considering:

$$v = A \sum_i (v \cdot b_i) b_i \tag{23}$$

$$= A \sum_i (b_i \otimes b_i) v. \tag{24}$$

Requiring that Equation 24 holds for any $v \in \mathbb{R}^d$, one can conclude $A = \sum_i (b_i \otimes b_i)^{-1}$. Finally, we obtain

$$v = [b_i \otimes b_i]^{-1} \sum_i (v \cdot b_i) b_i \tag{25}$$

For more details on frames, see, e.g., Han et al. (2007).

Then, we can derive NIsoGCN at the $i$th vertex on the Neumann boundary, by letting

$$B = \left\{ \frac{x_{j_1} - x_i}{\|x_{j_1} - x_i\|}, \frac{x_{j_2} - x_i}{\|x_{j_2} - x_i\|}, \ldots, \frac{x_{j_m} - x_i}{\|x_{j_m} - x_i\|}, \sqrt{w_i} n_i \right\}, \tag{26}$$

where $\{j_1, j_2, \ldots, j_m\}$ are indices of neighboring vertices to the $i$th vertex. In addition, we assume the approximated gradient of a scalar field $\psi$ at the $i$th vertex, $\langle \nabla \psi \rangle_i$, satisfies the following conditions:

$$\langle \nabla \psi \rangle_i \cdot \frac{x_{j_k} - x_i}{\|x_{j_k} - x_i\|} = \frac{\psi_{j_k} - \psi_i}{\|x_{j_k} - x_i\|}, \qquad (k = 1, \ldots, m), \tag{27}$$

$$\langle \nabla \psi \rangle_i \cdot n = \hat{g}_i. \tag{28}$$

Equation 27 is a natural assumption because we expect the directional derivative in the direction of $(\boldsymbol{x}_{j_k} - \boldsymbol{x}_i)/\|\boldsymbol{x}_{j_k} - \boldsymbol{x}_i\|$ should correspond to the slope of $\psi$ in the same direction. Equation 28 is the Neumann boundary condition, which we want to satisfy. Finally, by substituting Equations 26, 27, and 28, we obtain NIsoGCN, i.e., Equation 14.

To apply NIsoGCN to $\boldsymbol{t}$, the rank $k$ tensors ($k \geq 1$), one can recursively define the operation as:

$$\text{NIsoGCN}_{k \to k+1}(\boldsymbol{t}) := \begin{pmatrix} \text{NIsoGCN}_{k-1 \to k}(\boldsymbol{t}_1) \\ \text{NIsoGCN}_{k-1 \to k}(\boldsymbol{t}_2) \\ \text{NIsoGCN}_{k-1 \to k}(\boldsymbol{t}_3) \end{pmatrix}, \tag{29}$$

where $\boldsymbol{t}_i$ is the $i$th component of $\boldsymbol{t}$, resulting in the rank $(k-1)$ tensor. In case of the rank 1 tensor $\boldsymbol{v}$, it can be formulated as:

$$\text{NIsoGCN}_{1 \to 2}(\boldsymbol{v}) := \begin{pmatrix} \text{NIsoGCN}_{0 \to 1}(v_1) \\ \text{NIsoGCN}_{0 \to 1}(v_2) \\ \text{NIsoGCN}_{0 \to 1}(v_3) \end{pmatrix} \approx \begin{pmatrix} \partial v_1/\partial x & \partial v_1/\partial y & \partial v_1/\partial z \\ \partial v_2/\partial x & \partial v_2/\partial y & \partial v_2/\partial z \\ \partial v_3/\partial x & \partial v_3/\partial y & \partial v_3/\partial z \end{pmatrix} = \nabla \boldsymbol{v}. \tag{30}$$

Please note that each component $v_i$ has multiple features in the encoded space, e.g., 16 or 64, resulting in $\text{NIsoGCN}_{1 \to 2}(\boldsymbol{v})$ represents multiple rank 2 tensors for each vertex (see Figure 1 of Horie et al. (2021)).

As discussed in Horie et al. (2021), IsoGCNs (NIsoGCNs) correspond to spatial differential operators as:

- $\text{NIsoGCN}_{0 \to 1}(\psi)$: Gradient $\nabla \psi$ (rank 0 tensor to rank 1 tensor)
- $\text{NIsoGCN}_{1 \to 0}(\boldsymbol{v})$: Divergence $\nabla \cdot \boldsymbol{v}$ (rank 1 tensor to rank 0 tensor)
- $\text{NIsoGCN}_{0 \to 1 \to 0}(\psi) := \text{NIsoGCN}_{1 \to 0} \circ \text{NIsoGCN}_{0 \to 1}(\psi)$: Laplacian $\nabla \cdot \nabla \psi$ (rank 0 tensor to rank 1 tensor to rank 0 tensor)
- $\text{NIsoGCN}_{1 \to 2}(\boldsymbol{v})$: Jacobian $\nabla \boldsymbol{v}$ (rank 1 tensor to rank 2 tensor)
- $\text{NIsoGCN}_{0 \to 1 \to 2}(\psi) := \text{NIsoGCN}_{l \to 2} \circ \text{NIsoGCN}_{0 \to 1}(\psi)$: Hessian $\nabla \nabla \psi$ (rank 0 tensor to rank 1 tensor to rank 2 tensor)

Because NIsoGCN contains a learnable weight matrix (see Equation 14), the component learns to predict the derivative of the corresponding tensor rank in an encoded space. This feature of NIsoGCNs enables us to construct machine learning models corresponding to PDE in the encoded space.

## A.3 Derivation of the step size in the Barzilai–Borwein method

We derive Equation 16 by applying the Barzilai–Borwein method to our case. We start with Equation 8, which corresponds to a nonlinear problem:

$$\boldsymbol{R}(\boldsymbol{v}) := \boldsymbol{v} - \boldsymbol{u}(t, \cdot) - \mathcal{D}(\boldsymbol{v})\Delta t, \tag{31}$$

$$\text{Solve}_{\boldsymbol{v}} \ \boldsymbol{R}(\boldsymbol{v})(\boldsymbol{x}_i) = \boldsymbol{0}, \ \forall i, \tag{32}$$

We consider solving it by applying the linear iterative method using the Taylor expansion, assuming the update $\Delta \boldsymbol{v}^{(i)} := \boldsymbol{v}^{(i+1)} - \boldsymbol{v}^{(i)}$ is small enough. The iterative method is expressed as:

$$\boldsymbol{v}^{(0)} = \boldsymbol{u}(t, \cdot), \tag{33}$$

$$\boldsymbol{v}^{(i+1)} = \boldsymbol{v}^{(i)} + \Delta \boldsymbol{v}^{(i)}, \tag{34}$$

$$\boldsymbol{R}(\boldsymbol{v}^{(i)} + \Delta \boldsymbol{v}^{(i)}) \approx \boldsymbol{R}(\boldsymbol{v}^{(i)}) + \nabla_{\boldsymbol{v}} \boldsymbol{R}(\boldsymbol{v}^{(i)}) \Delta \boldsymbol{v}^{(i)} = \boldsymbol{0}, \tag{35}$$

where $\nabla_{\boldsymbol{v}} \boldsymbol{R}(\boldsymbol{v}^{(i)})$ denotes the Jacobian matrix with the shape of $n \times n$ ($n$ roughly corresponds to the number of vertices of the mesh). To optain update, we may solve Equation 35 as:

$$\Delta \boldsymbol{v}^{(i)} = \left[\nabla_{\boldsymbol{v}} \boldsymbol{R}(\boldsymbol{v}^{(i)})\right]^{-1} \boldsymbol{R}(\boldsymbol{v}^{(i)}), \tag{36}$$

corresponding to the Newton–Raphson method. However, it may take enormous computation resources because $\nabla_{\boldsymbol{v}} \boldsymbol{R}(\boldsymbol{v}^{(i)})$ is usually a huge matrix. Instead, we can approximate:

$$\left[\nabla_{\boldsymbol{v}} \boldsymbol{R}(\boldsymbol{v}^{(i)})\right]^{-1} \approx \alpha^{(i)}, \tag{37}$$

which corresponds to gradient descent:

$$\Delta \boldsymbol{v}^{(i)} \approx \alpha^{(i)} \boldsymbol{R}(\boldsymbol{v}^{(i)}). \tag{38}$$

Substituting Equation 37 into Equation 35, we obtain:

$$\boldsymbol{R}(\boldsymbol{v}^{(i+1)}) \approx \boldsymbol{R}(\boldsymbol{v}^{(i)}) + \frac{1}{\alpha^{(i)}} \Delta \boldsymbol{v}^{(i)}. \tag{39}$$

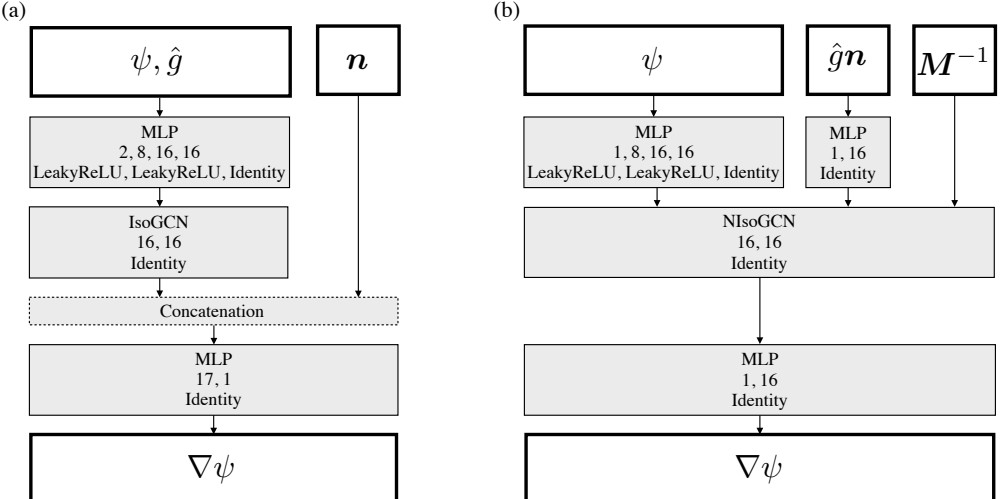

Figure 5: Architecture used for (a) original IsoGCN and (b) NIsoGCN training. In each cell, we put the number of units in each layer along with the activation functions used.

We want to find a good $\alpha^{(i)}$ satisfying Equation 39 the best. Thus, we obtain $\alpha_{\mathrm{BB}}^{(i)}$ through:

$$\alpha_{\mathrm{BB}}^{(i)} = \arg\min_{\alpha} \mathcal{L}(\alpha), \tag{40}$$

$$\mathbb{R} \ni \mathcal{L}(\alpha) := \frac{1}{2}\left\|\Delta\boldsymbol{v}^{(i)} - \alpha\Delta\boldsymbol{R}^{(i)}\right\|^2, \text{ where } \Delta\boldsymbol{R}^{(i)} = \boldsymbol{R}(\boldsymbol{v}^{(i+1)}) - \boldsymbol{R}(\boldsymbol{v}^{(i)}). \tag{41}$$

Because of the convexity of the problem, it is enough to find alpha satisfying:

$$\left.\frac{\mathrm{d}\mathcal{L}}{\mathrm{d}\alpha}\right|_{\alpha_{\mathrm{BB}}^{(i)}} = \left\langle \Delta\boldsymbol{v}^{(i)} - \alpha_{\mathrm{BB}}^{(i)}\Delta\boldsymbol{R}^{(i)}, -\Delta\boldsymbol{R}^{(i)} \right\rangle = 0, \tag{42}$$

where $< \cdot, \cdot >$ denotes the inner product in the corresponding space. Using the linearity of the inner product, we obtain:

$$\left\langle \Delta\boldsymbol{v}^{(i)} - \alpha_{\mathrm{BB}}^{(i)}\Delta\boldsymbol{R}^{(i)}, -\Delta\boldsymbol{R}^{(i)} \right\rangle = 0, \tag{43}$$

$$-\left\langle \Delta\boldsymbol{v}^{(i)}, \Delta\boldsymbol{R}^{(i)} \right\rangle + \alpha_{\mathrm{BB}}^{(i)} \left\langle \Delta\boldsymbol{R}^{(i)}, \Delta\boldsymbol{R}^{(i)} \right\rangle = 0, \tag{44}$$

$$\alpha_{\mathrm{BB}}^{(i)} = \frac{\left\langle \Delta\boldsymbol{v}^{(i)}, \Delta\boldsymbol{R}^{(i)} \right\rangle}{\left\langle \Delta\boldsymbol{R}^{(i)}, \Delta\boldsymbol{R}^{(i)} \right\rangle}. \tag{45}$$

Equation 45 is equivalent to Equation 16.

As seen from the derivation, $\alpha_{\mathrm{BB}}^{(i)}$ is determined to satisfy Equation 39 as much as possible for all vertices and all feature components. That means $\alpha_{\mathrm{BB}}^{(i)}$ has global information because it considers all vertices, making the global interaction possible. In addition, $\alpha_{\mathrm{BB}}^{(i)}$ is equivariant because it is scalar, which does not depend on coordinate. Therefore, $\alpha_{\mathrm{BB}}^{(i)}$ is suitable for realizing efficient PDE solvers with $\mathrm{E}(n)$-equivariance.

## B Experiment details: gradient dataset

Figure 5 shows the architectures we used for the gradient dataset. The dataset is uploaded online.[6] We followed the instruction of Horie et al. (2021) (in particular, Appendix D.1 of their paper) to make the features and models equivariant. To facilitate a fair comparison, we made input information for both models equivalent, except for $\boldsymbol{M}^{-1}$ in Equation Equation 15, which is a part of our novelty. For both models, we used Adam (Kingma & Ba, 2014) as an optimizer with the default setting. Training for both models took around ten minutes using one GPU (NVIDIA A100 for NVLink 40GiB HBM2). Figure 5 shows model architectures used for the experiment.

---

[6] https://savanna.ritc.jp/~horiem/penn_neurips2022/data/grad/grad_data.tar.gz

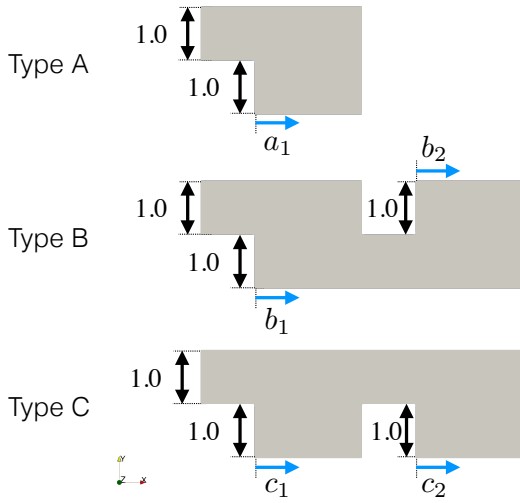

Figure 6: Three template shapes used to generate the dataset. $a_1$, $b_1$, $b_2$, $c_1$, and $c_2$ are the design parameters.

## C Experiment details: incompressible flow dataset

### C.1 Governing equation

The incompressible Navier–Stokes equations, the governing equations of incompressible flow, are expressed as:

$$\frac{\partial \boldsymbol{u}}{\partial t} = -(\boldsymbol{u} \cdot \nabla)\boldsymbol{u} + \frac{1}{\text{Re}} \nabla \cdot \nabla \boldsymbol{u} - \nabla p \qquad (t, \boldsymbol{x}) \in (0, T) \times \Omega, \qquad (46)$$

$$\boldsymbol{u} = \hat{\boldsymbol{u}} \qquad (t, \boldsymbol{x}) \in \partial\Omega_{\text{Dirichlet}}^{(\boldsymbol{u})}, \qquad (47)$$

$$\left[ \nabla \boldsymbol{u} + (\nabla \boldsymbol{u})^T \right] \boldsymbol{n} = \boldsymbol{0} \qquad (t, \boldsymbol{x}) \in \partial\Omega_{\text{Neumann}}^{(\boldsymbol{u})}. \qquad (48)$$

We also consider the following incompressible condition:

$$\nabla \cdot \boldsymbol{u} = 0 \qquad (t, \boldsymbol{x}) \in (0, T) \times \Omega, \qquad (49)$$

which may be problematic when solving these equations numerically. Therefore, it is common to divide the equations into two: one to obtain pressure and one to compute velocity. There are many methods to make such a division; for instance, the fractional step method derives the Poisson equation for pressure as follows:

$$\nabla \cdot \nabla p(t + \Delta t, \boldsymbol{x}) = \frac{1}{\Delta t} (\nabla \cdot \tilde{\boldsymbol{u}})(t, \boldsymbol{x}), \qquad (50)$$

where

$$\tilde{\boldsymbol{u}} = \boldsymbol{u} - \Delta t \left( \boldsymbol{u} \cdot \nabla \boldsymbol{u} - \frac{1}{\text{Re}} \nabla \cdot \nabla \boldsymbol{u} \right) \qquad (51)$$

is called the intermediate velocity. Once we solve the equation, we can compute the time evolution of velocity as follows:

$$\boldsymbol{u}(t + \Delta t, \boldsymbol{x}) = \tilde{\boldsymbol{u}}(t, \boldsymbol{x}) - \Delta t \nabla p(t + \Delta t, \boldsymbol{x}). \qquad (52)$$

Because the fractional step method requires solving the Poisson equation for pressure, we also need the boundary conditions for pressure as well:

$$p = 0 \qquad (t, \boldsymbol{x}) \in \partial\Omega_{\text{Dirichlet}}^{(p)}, \qquad (53)$$

$$\nabla p \cdot \boldsymbol{n} = 0 \qquad (t, \boldsymbol{x}) \in \partial\Omega_{\text{Neumann}}^{(p)}. \qquad (54)$$

Our machine learning task is also based on the same assumption: motivating pressure prediction in addition to velocity with boundary conditions of both.

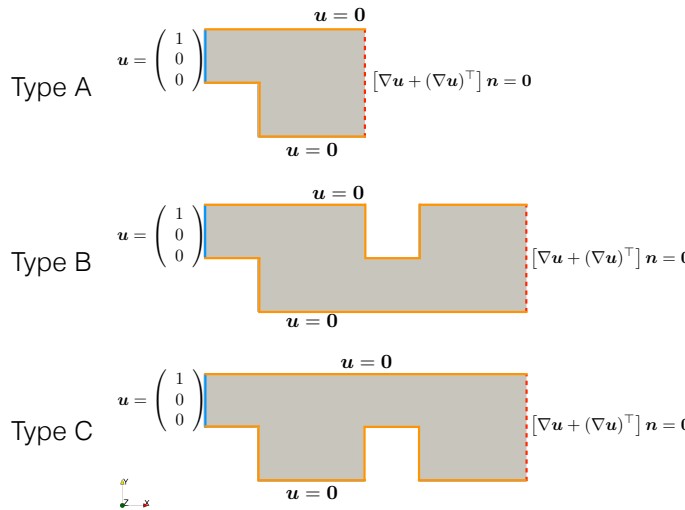

Figure 7: Boundary conditions of $\boldsymbol{u}$ used to generate the dataset. The continuous lines and dotted lines correspond to Dirichlet and Neumann boundaries.

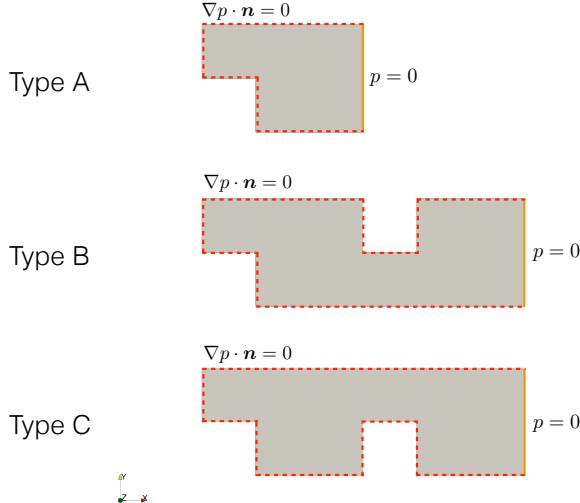

Figure 8: Boundary conditions of $p$ used to generate the dataset. The continuous lines and dotted lines correspond to Dirichlet and Neumann boundaries.

## C.2 Dataset

We generated numerical analysis results using various shapes of the computational domain, starting from three template shapes and changing their design parameters as shown in Figure 6. For each design parameter, we varied from 0 to 1.0 with a step size of 0.1, yielding 11 shapes for type A and 121 shapes for type B and C. The boundary conditions were set as shown in Figures 7 and 8. These design and boundary conditions were chosen to have the characteristic length of 1.0 and flow speed of 1.0. The viscosity was set to $10^{-3}$, resulting in Reynolds number $\mathrm{Re} \sim 10^3$. The linear solvers used were generalized geometric-algebraic multi-grid for $p$ and the smooth solver with the Gauss–Siedel smoother for $\boldsymbol{u}$. Numerical analysis to generate each sample took up to one hour using CPU one core (Intel Xeon CPU E5-2695 v2@2.40GHz). The dataset is uploaded online.[7]

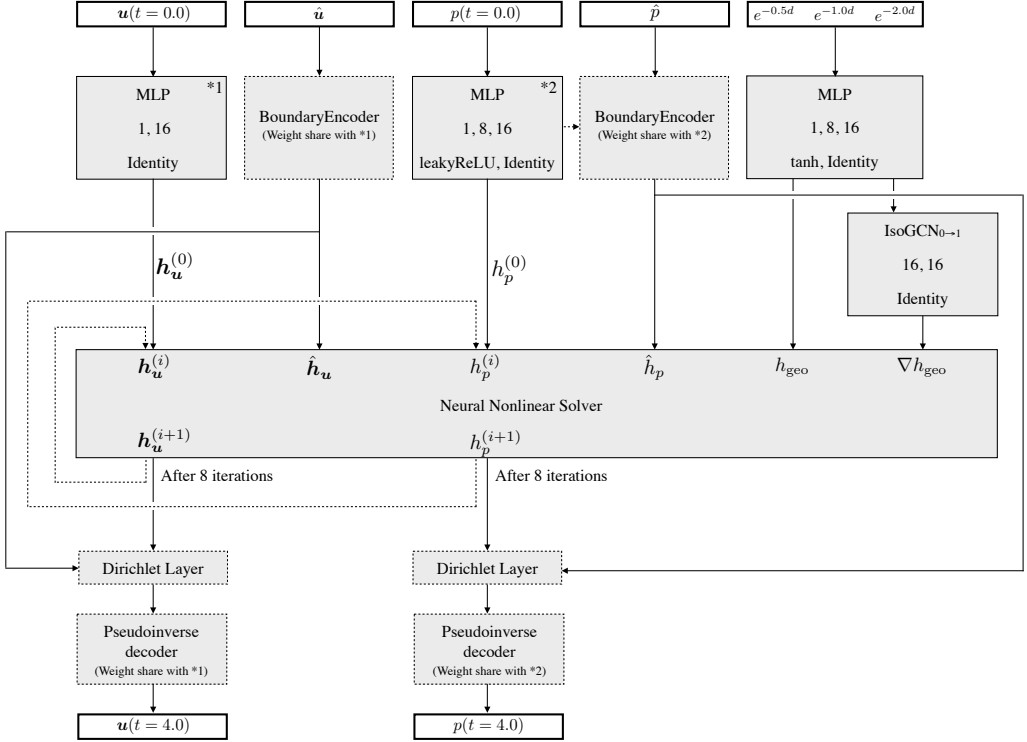

Figure 9: The overview of the PENN architecture for the incompressible flow dataset. Gray boxes with continuous (dotted) lines are trainable (untrainable) components. Arrows with dotted lines correspond to the loop. In each cell, we put the number of units in each layer along with the activation functions used.

## C.3 Model architectures

The input features of the model are:

- $\boldsymbol{u}(t = 0.0)$: The initial velocity field, the solulsion of potential flow
- $\hat{\boldsymbol{u}}$: The Dirichlet boundary condition for velocity
- $p(t = 0.0)$: The initial pressure field
- $\hat{p}$: The Dirichlet boundary condition for pressure
- $e^{-0.5d}, e^{-1.0d}, e^{-2.0d}$: Features computed from $d$, the distance from the wall boundary condition

and the output features are:

- $\boldsymbol{u}(t = 4.0)$: The velocity field at $t = 4.0$
- $p(t = 4.0)$: The pressure field at $t = 4.0$

The strategy to construct PENN for the incompressible flow dataset is the following:

- Consider the encoded version of the governing equation
- Apply the neural nonlinear solver containing the Dirichlet layer and the NIsoGCN to the encoded equation
- Decode the hidden feature using the pseudoinverse decoder.

---

[7] https://savanna.ritc.jp/~horiem/penn_neurips2022/data/fluid/fluid_data.tar.gz. parta[a-e]

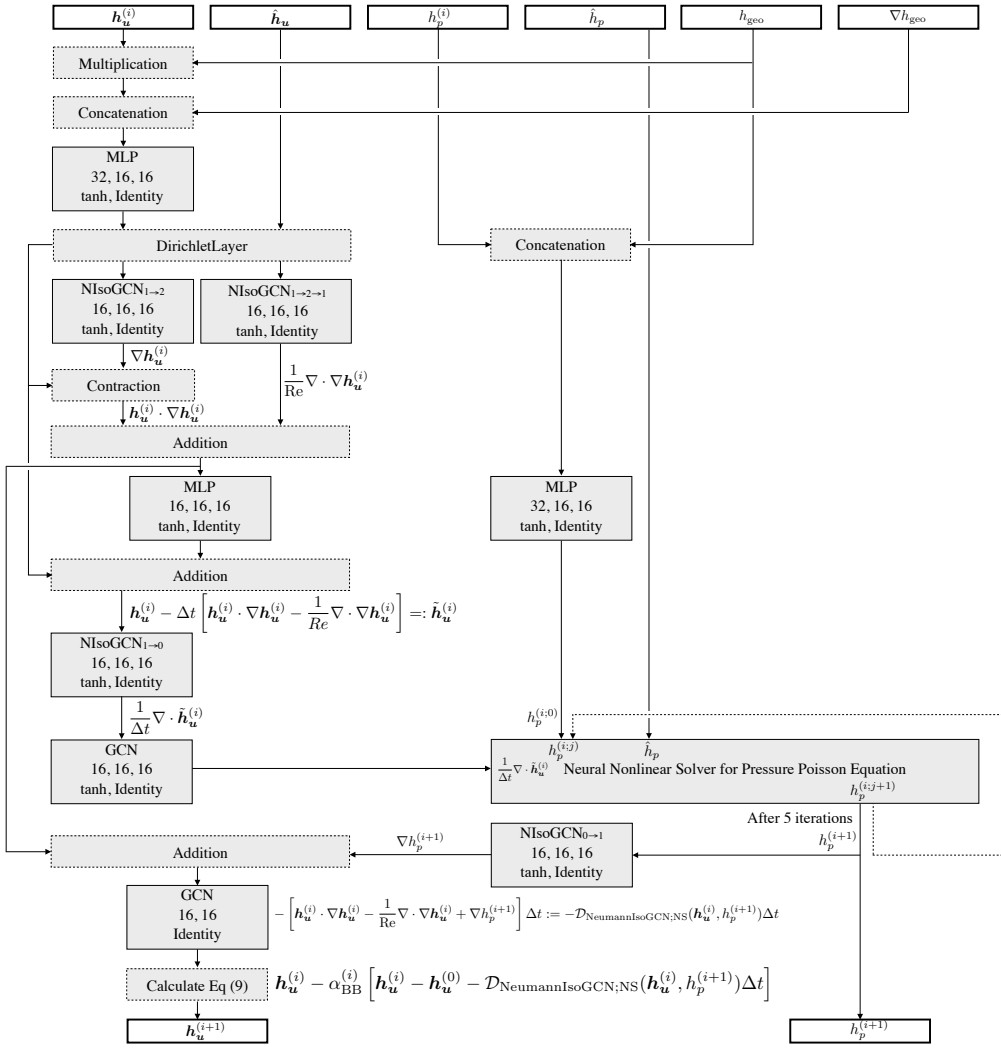

Figure 10: The neural nonlinear solver for velocity. Gray boxes with continuous (dotted) lines are trainable (untrainable) components. Arrows with dotted lines correspond to the loop. In each cell, we put the number of units in each layer along with the activation functions used.

Reflecting the fractional step method, we build PENN using spatial differential operators provided by NIsoGCN. We use a simple linear encoder for the velocity and the associated Dirichlet boundary conditions. For pressure and its Dirichlet constraint, we use a simple MLP with one hidden layer. We encode each feature in a 16-dimensional space. After features are encoded, we apply a neural nonlinear solver containing NeumanIsoGCNs and Dirichlet layers, reflecting the fractional step method (Equations 51 and 52).

The encoded equations are expressed as:

$$[\text{NIsoGCN}_{1\to0} \circ \text{NIsoGCN}_{0\to1}(h_p)](t + \Delta t, \boldsymbol{x}) = \frac{1}{\Delta t} \left[ \text{NIsoGCN}_{1\to0}\left(\tilde{\boldsymbol{h}}_{\boldsymbol{u}}\right) \right](t, \boldsymbol{x}), \tag{55}$$

$$\tilde{\boldsymbol{h}}_{\boldsymbol{u}} := \boldsymbol{h}_{\boldsymbol{u}} - \Delta t \left[ \boldsymbol{h}_{\boldsymbol{u}} \cdot \text{NIsoGCN}_{1\to2}\left(\boldsymbol{h}_{\boldsymbol{u}}\right) - \frac{1}{\text{Re}} \text{NIsoGCN}_{2\to1} \circ \text{NIsoGCN}_{1\to2}\left(\boldsymbol{h}_{\boldsymbol{u}}\right) \right], \tag{56}$$

$$\boldsymbol{h}_{\boldsymbol{u}}(t + \Delta t, \boldsymbol{x}) = \tilde{\boldsymbol{h}}_{\boldsymbol{u}}(t, \boldsymbol{x}) - \Delta t\, \text{NIsoGCN}_{0\to1}(h_p)(t + \Delta t, \boldsymbol{x}), \tag{57}$$

where $\boldsymbol{h}_{\boldsymbol{u}}$ is the encoded $\boldsymbol{u}$ and $h_p$ is the encoded $p$. Note that these equations correspond to Equations 50, 51, and 52, by regarding IsoGCNs as spatial derivative operators. The corresponding neural nonlinear solvers are

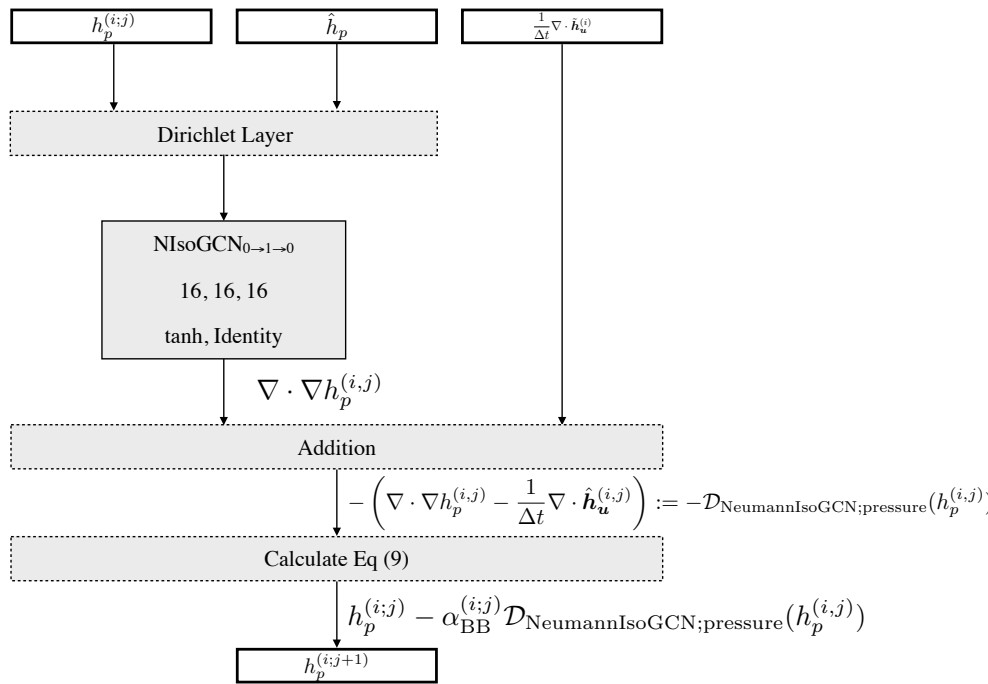

Figure 11: The neural nonlinear solver for pressure. Gray boxes with continuous (dotted) lines are trainable (untrainable) components. In each cell, we put the number of units in each layer along with the activation functions used.

expressed as:

$$\boldsymbol{h}_{\boldsymbol{u}}^{(i+1)} = \boldsymbol{h}_{\boldsymbol{u}}^{(i)} - \alpha_{\mathrm{BB}}^{(i)} \left[ \boldsymbol{h}_{\boldsymbol{u}}^{(i)} - \boldsymbol{h}_{\boldsymbol{u}}^{(0)} - \mathcal{D}_{\mathrm{NIsoGCN;NS}} \left( \boldsymbol{h}_{\boldsymbol{u}}^{(i)}, h_{p}^{(i+1)} \right) \Delta t \right], \tag{58}$$

$$\mathcal{D}_{\mathrm{NIsoGCN;NS}} \left( \boldsymbol{h}_{\boldsymbol{u}}^{(i)}, h_{p}^{(i+1)} \right)$$
$$:= \left[ \boldsymbol{h}_{\boldsymbol{u}}^{(i)} \cdot \mathrm{NIsoGCN}_{1\to2} \left( \boldsymbol{h}_{\boldsymbol{u}}^{(i)} \right) - \frac{1}{\mathrm{Re}} \mathrm{NIsoGCN}_{2\to1} \circ \mathrm{NIsoGCN}_{1\to2} \left( \boldsymbol{h}_{\boldsymbol{u}}^{(i)} \right) + \mathrm{NIsoGCN} \left( h_{p}^{(i+1)} \right) \right], \tag{59}$$

for $\boldsymbol{h}_{\boldsymbol{u}}$ and

$$h_{p}^{(i;j+1)} = h_{p}^{(i;j)} - \alpha_{\mathrm{BB}}^{(i;j)} \mathcal{D}_{\mathrm{NIsoGCN;pressure}}(h_{p}^{(i;j)}), \tag{60}$$

$$\mathcal{D}_{\mathrm{NIsoGCN;pressure}} \left( h_{p}^{(i;j)} \right) := \left( \mathrm{NIsoGCN}_{1\to0} \circ \mathrm{NIsoGCN}_{0\to1} \left( h_{p}^{(\cdot;j)} \right) - \frac{1}{\Delta t} \mathrm{NIsoGCN}_{1\to0} \left( \hat{\boldsymbol{h}}_{\boldsymbol{u}}^{(i)} \right) \right), \tag{61}$$

for $h_p$, where $\boldsymbol{h}_{\boldsymbol{u}}^{(0)} = \boldsymbol{h}_{\boldsymbol{u}}(t, \cdot)$, $h_{p}^{(0)} = h_p(t, \cdot)$, and $h_{p}^{(i;0)} = h_{p}^{(i)}$. For notation regarding NIsoGCNs, please see Appendix A.2. Figures 9, 10, and 11 present the PENN model architecture used for the incompressible flow dataset.

As seen in Figure 10, we have a subloop that solves the Poisson equation for pressure in the nonlinear solver's loop for velocity. We looped the solver for pressure five times and eight times for velocity. After these loops stopped, we decoded the hidden features to obtain predictions for velocity and pressure, using the corresponding pseudoinverse decoders.

## C.4 Implementation details

As discussed in Horie et al. (2021), nonlinearity can be applied to the scalar but cannot be applied to the tensors with a rank equal to or greater than one. For such a tensor, nonlinearity can be applied to its norm as:

$$\mathrm{MLP}_{\mathrm{tensor}}(\boldsymbol{v}) := \mathrm{MLP}(\|\boldsymbol{v}\|)\boldsymbol{v}. \tag{62}$$

This strategy to apply nonlinearity is used not only in the MLP blocks but also NIsoGCN blocks. To facilitate the smoothness of pressure and velocity fields, we apply GCN layers corresponding to numerical viscosity in the standard numerical analysis method. Here, please note that the PENN model consists of components that accept arbitrary input lengths, e.g., pointwise MLPs, deep sets, and NIsoGCNs. Thanks to the model's flexibility, we can apply the same model to arbitrary meshes similar to other GNNs.

## C.5 Training details

Because the neural nonlinear solver applies the same layers many times during the loop, the model behaved somehow similar to recurrent neural networks during training, which could cause instability. To avoid such unwanted behavior, we simply retried training by reducing the learning rate of the Adam optimizer by a factor of 0.5. We found our way of training useful compared to using the learning rate schedule because sometimes the loss value of PENN can be extremely high, resulting in difficulty to reach convergence with a lower learning rate after such an explosion. Therefore, we applied early stopping and restarted training using a lower learning rate from the epoch with the best validation loss. Our initial learning rate was $5.0 \times 10^{-4}$, and we restarted the training twice, which was done automatically, within the 24-hour training period of PENN. For the ablation study, we used the same setting for all models. For PENN and ablation models, we used Adam (Kingma & Ba, 2014) as an optimizer. For MP-PDE solvers, we used the default setting written in the paper and the code.

## C.6 Result details

Table 4 presents the detailed results of the comparison between MP-PDE and PENN. Interestingly, the performance of MP-PDE gets better as the time window size increases. Therefore, our future direction may be to incorporate MP-PDE's temporal bundling and pushforward trick into PENN to enable us to predict the state after a far longer time than we do in the present work.

Tables 5 and 6 show the speed and accuracy of the machine learning models tested. PENN models show excellent performance with a lot smaller number of parameters compared to MP-PDE models. It is achieved due to efficient parameter sharing in the proposed model, e.g., the same weights are used repeatedly in the neural nonlinear encoder. Also, as pointed out in Ravanbakhsh et al. (2017), there is a strong connection between parameter sharing and equivariance. PENN has equivariance in, e.g., permutation, time translation, and $\mathrm{E}(n)$ through parameter sharing, which is in line with them.

Table 7 presents the speed and accuracy with various settings of OpenFOAM to seek a speed-accuracy tradeoff. We tested three configurations of linear solvers:

- Generalized geometric-algebraic multi-grid (GAMG) for $p$ and the smooth solver for $\boldsymbol{u}$
- Generalized geometric-algebraic multi-grid (GAMG) for both $p$ and $\boldsymbol{u}$
- The smooth solver for $p$ and $\boldsymbol{u}$

In addition, we tested different resolutions for space and time by changing:

- The number of divisions per unit length: 22.5, 45.0, 90.0
- Time step size: 0.001, 0.005, 0.010, 0.050

Ground truth is computed using the number of divisions per unit length of 90.0 and time step size of 0.001; thus, this combination is eliminated from the comparison because the MSE error is underestimated (in particular, zero).

## C.7 Ablation study details

To validate the effectiveness of our model through an ablation study on the following settings:

(A) Without encoded boundary: In the nonlinear loop, we decode features to apply boundary conditions to fulfill Dirichlet conditions in the original physical space

(B) Without boundary condition in the neural nonlinear solver: We removed the Dirichlet layer in the nonlinear loop. Instead, we added the Dirichlet layer after the (non-pseudoinverse) decoder.

(C) Without neural nonlinear solver: We removed the nonlinear solver from the model and used the explicit time-stepping instead

(D) Without boundary condition input: We removed the boundary condition from input features

(E) Without Dirichlet layer: We removed the Dirichlet layer. Instead, we let the model learn to satisfy boundary conditions during training.

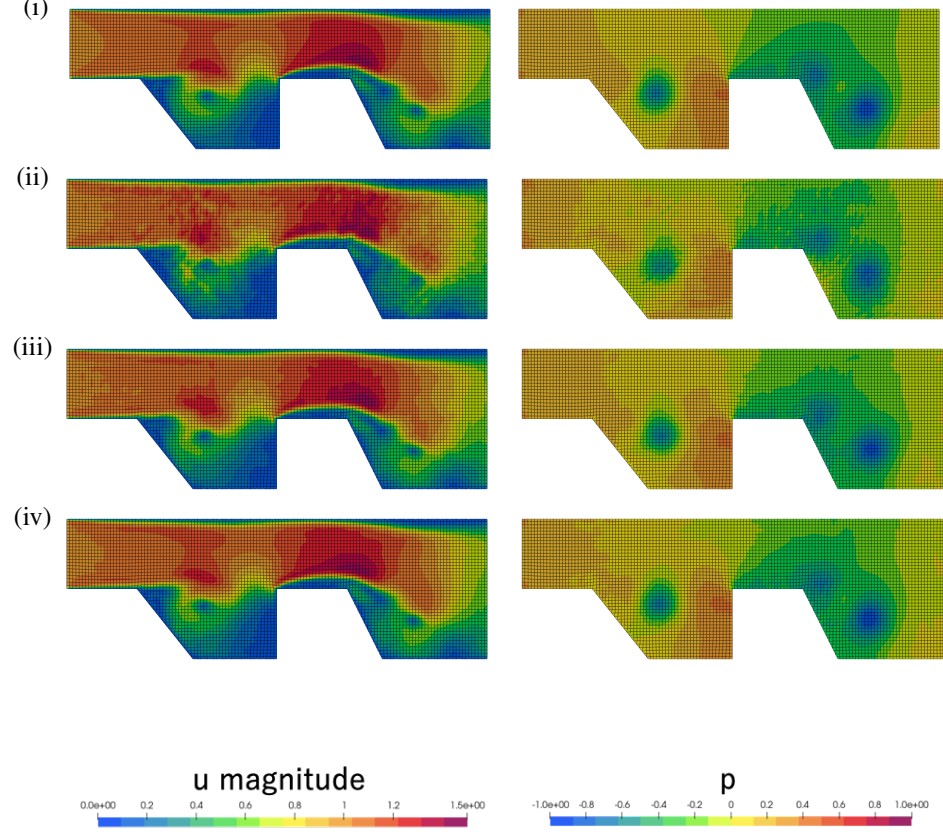

Figure 12: Visual comparison of the ablation study of (i) ground truth, (ii) the model without the neural nonlinear solver (Model (C)), (iii) the model without pseudoinverse decoder with Dirichlet layer after decoding (Model (G)), and (iv) PENN. It can be observed that PENN improves the prediction smoothness, especially for the velocity field.

(F) Without pseudoinverse decoder: We removed the pseudoinverse decoder and used simple MLPs for decoders.

(G) Without pseudoinverse decoder with Dirichlet boundary layer after decoding: Same as above, but with Dirichlet layer after decoding.

We again put the results of the ablation study in Table 8, which is already presented in Table 3, for the convenience of the readers.

Comparison with Model (A) shows that the nonlinear loop in the encoded space is inevitable for machine learning. This result is quite convincing because if the loop is made in the original space, the advantage of the expressive power of the neural networks cannot be leveraged. Comparison with Model (C) confirms that the concept of the solver is effective compared to simply stacking GNNs, corresponding to the explicit method.

If the boundary condition input is excluded (Model (D)), the performance degrades in line with Brandstetter et al. (2022). That model also has an error on the Dirichlet boundaries. Model (E) shows a similar result, improving performance using the information of the boundary conditions. If the pseudoinverse decoder is excluded (Model (F)), the output may not satisfy the Dirichlet boundary conditions as well. Besides, the decoder has more effect than expected because PENN is better than Model (G). Both models satisfy the Dirichlet boundary condition, while PENN has significant improvement. This may be because the pseudoinverse decoder facilitates the spatial continuity of the outputs in addition to the fulfillment of the Dirichlet boundary condition. In other words, using a simple decoder and the Dirichlet layer after that may cause spatial discontinuity of outputs. Visual comparison of part of the ablation study is shown in Figure 12.

Table 4: MSE loss ($\pm$ the standard error of the mean) on test dataset of incompressible flow. If "Trans." is "Yes", it means evaluation on randomly rotated and transformed test dataset. $n$ denotes the number of hidden features, $r$ denotes the number of iterations in the neural nonlinear solver used in PENN models, and TW denotes the time window size used in MP-PDE models.

| Method | Trans. | $\boldsymbol{u}$ $(\times 10^{-4})$ | $p$ $(\times 10^{-3})$ | $\hat{\boldsymbol{u}}_{\text{Dirichlet}}$ $(\times 10^{-4})$ | $\hat{p}_{\text{Dirichlet}}$ $(\times 10^{-3})$ |
|---|---|---|---|---|---|
| PENN $n = 16, r = 8$ | No | $4.36 \pm 0.03$ | $1.17 \pm 0.01$ | $0.00 \pm 0.00$ | $0.00 \pm 0.00$ |
| | Yes | $4.36 \pm 0.03$ | $1.17 \pm 0.01$ | $0.00 \pm 0.00$ | $0.00 \pm 0.00$ |
| PENN $n = 16, r = 4$ | No | $29.09 \pm 0.17$ | $11.35 \pm 0.04$ | $0.00 \pm 0.00$ | $0.00 \pm 0.00$ |
| | Yes | $29.09 \pm 0.17$ | $11.35 \pm 0.04$ | $0.00 \pm 0.00$ | $0.00 \pm 0.00$ |
| PENN $n = 8, r = 8$ | No | $177.42 \pm 0.93$ | $35.70 \pm 0.12$ | $0.00 \pm 0.00$ | $0.00 \pm 0.00$ |
| | Yes | $177.42 \pm 0.93$ | $35.70 \pm 0.12$ | $0.00 \pm 0.00$ | $0.00 \pm 0.00$ |
| PENN $n = 8, r = 4$ | No | $26.82 \pm 0.16$ | $7.86 \pm 0.03$ | $0.00 \pm 0.00$ | $0.00 \pm 0.00$ |
| | Yes | $26.82 \pm 0.16$ | $7.86 \pm 0.03$ | $0.00 \pm 0.00$ | $0.00 \pm 0.00$ |
| PENN $n = 4, r = 8$ | No | $92.80 \pm 0.52$ | $31.47 \pm 0.13$ | $0.00 \pm 0.00$ | $0.00 \pm 0.00$ |
| | Yes | $92.80 \pm 0.52$ | $31.47 \pm 0.13$ | $0.00 \pm 0.00$ | $0.00 \pm 0.00$ |
| PENN $n = 4, r = 4$ | No | $120.35 \pm 0.65$ | $35.53 \pm 0.12$ | $0.00 \pm 0.00$ | $0.00 \pm 0.00$ |
| | Yes | $120.35 \pm 0.65$ | $35.53 \pm 0.12$ | $0.00 \pm 0.00$ | $0.00 \pm 0.00$ |
| MP-PDE $n = 128, \text{TW} = 20$ | No | $1.30 \pm 0.01$ | $1.32 \pm 0.01$ | $0.45 \pm 0.01$ | $0.28 \pm 0.02$ |
| | Yes | $1953.62 \pm 7.62$ | $281.86 \pm 0.78$ | $924.73 \pm 6.14$ | $202.97 \pm 3.81$ |
| MP-PDE $n = 128, \text{TW} = 10$ | No | $12.08 \pm 0.11$ | $6.49 \pm 0.03$ | $1.36 \pm 0.01$ | $2.57 \pm 0.05$ |
| | Yes | $1468.12 \pm 5.75$ | $192.97 \pm 0.57$ | $767.17 \pm 4.36$ | $51.87 \pm 1.07$ |
| MP-PDE $n = 128, \text{TW} = 4$ | No | $32.07 \pm 0.33$ | $6.22 \pm 0.05$ | $0.85 \pm 0.01$ | $0.92 \pm 0.03$ |
| | Yes | $2068.99 \pm 8.30$ | $180.54 \pm 0.57$ | $284.72 \pm 1.69$ | $59.21 \pm 1.32$ |
| MP-PDE $n = 128, \text{TW} = 2$ | No | $58.88 \pm 0.60$ | $9.62 \pm 0.07$ | $1.02 \pm 0.02$ | $2.83 \pm 0.10$ |
| | Yes | $1853.27 \pm 7.89$ | $219.59 \pm 0.53$ | $965.90 \pm 28.61$ | $358.53 \pm 2.13$ |
| MP-PDE $n = 64, \text{TW} = 20$ | No | $6.09 \pm 0.05$ | $5.39 \pm 0.03$ | $1.65 \pm 0.02$ | $2.16 \pm 0.08$ |
| | Yes | $1969.34 \pm 7.50$ | $388.54 \pm 1.12$ | $720.35 \pm 5.15$ | $218.06 \pm 8.01$ |
| MP-PDE $n = 64, \text{TW} = 10$ | No | $38.54 \pm 0.32$ | $31.33 \pm 0.09$ | $2.04 \pm 0.02$ | $5.87 \pm 0.09$ |
| | Yes | $2738.84 \pm 9.37$ | $171.32 \pm 0.60$ | $417.57 \pm 2.49$ | $28.34 \pm 0.92$ |
| MP-PDE $n = 64, \text{TW} = 2$ | No | $125.09 \pm 1.11$ | $21.93 \pm 0.09$ | $2.27 \pm 0.03$ | $5.92 \pm 0.16$ |
| | Yes | $1402.01 \pm 6.03$ | $435.75 \pm 2.41$ | $384.30 \pm 4.13$ | $57.26 \pm 1.90$ |
| MP-PDE $n = 32, \text{TW} = 20$ | No | $32.46 \pm 0.24$ | $17.40 \pm 0.07$ | $5.92 \pm 0.05$ | $5.94 \pm 0.17$ |
| | Yes | $2201.16 \pm 7.59$ | $351.66 \pm 0.82$ | $429.30 \pm 3.27$ | $562.16 \pm 11.62$ |
| MP-PDE $n = 32, \text{TW} = 10$ | No | $115.30 \pm 1.01$ | $34.97 \pm 0.15$ | $10.26 \pm 0.09$ | $6.84 \pm 0.14$ |
| | Yes | $2824.76 \pm 8.60$ | $496.33 \pm 1.33$ | $2276.11 \pm 10.57$ | $488.50 \pm 5.01$ |
| MP-PDE $n = 32, \text{TW} = 4$ | No | $272.73 \pm 2.07$ | $94.27 \pm 0.45$ | $11.50 \pm 0.12$ | $35.76 \pm 0.29$ |
| | Yes | $1973.35 \pm 8.29$ | $554.69 \pm 4.26$ | $647.31 \pm 7.40$ | $157.85 \pm 8.41$ |
| MP-PDE $n = 32, \text{TW} = 2$ | No | $794.90 \pm 4.68$ | $82.61 \pm 0.40$ | $50.23 \pm 0.91$ | $31.41 \pm 1.88$ |
| | Yes | $3240.69 \pm 21.91$ | $443.10 \pm 2.56$ | $2885.30 \pm 41.17$ | $562.08 \pm 19.28$ |

Table 5: MSE loss ($\pm$ the standard error of the mean) of PENN models on test dataset of incompressible flow.

| # hidden feature | # iteration in the neural nonlinear solver | # parameter | Total MSE ($\times 10^{-3}$) | Total time [s] |
|---|---|---|---|---|
| 16 | 8 | 8,432 | $1.61 \pm 0.01$ | $5.33 \pm 0.13$ |
| 16 | 4 | 8,432 | $14.26 \pm 0.03$ | $2.52 \pm 0.06$ |
| 8 | 8 | 2,100 | $53.44 \pm 0.11$ | $3.54 \pm 0.08$ |
| 8 | 4 | 2,100 | $10.54 \pm 0.03$ | $2.16 \pm 0.04$ |
| 4 | 8 | 596 | $40.75 \pm 0.10$ | $2.86 \pm 0.06$ |
| 4 | 4 | 596 | $47.57 \pm 0.10$ | $1.35 \pm 0.04$ |

Table 6: MSE loss ($\pm$ the standard error of the mean) of MP-PDE models on test dataset of incompressible flow.

| # hidden feature | Time window size | # parameter | Total MSE ($\times 10^{-3}$) | Total MSE (Trans.) ($\times 10^{-3}$) | Total time [s] |
|---|---|---|---|---|---|
| 128 | 20 | 709,316 | $1.45 \pm 0.01$ | $477.23 \pm 0.77$ | $51.61 \pm 1.41$ |
| 128 | 10 | 673,484 | $7.70 \pm 0.02$ | $339.78 \pm 0.57$ | $94.01 \pm 2.66$ |
| 128 | 4 | 651,972 | $9.43 \pm 0.04$ | $387.44 \pm 0.71$ | $137.32 \pm 3.91$ |
| 128 | 2 | 644,548 | $15.51 \pm 0.07$ | $404.92 \pm 0.67$ | $57.28 \pm 1.91$ |
| 64 | 20 | 204,004 | $6.00 \pm 0.02$ | $585.48 \pm 0.95$ | $13.62 \pm 0.38$ |
| 64 | 10 | 185,356 | $35.19 \pm 0.07$ | $445.20 \pm 0.79$ | $23.73 \pm 0.67$ |
| 64 | 2 | 174,740 | $34.44 \pm 0.10$ | $575.95 \pm 1.76$ | $32.61 \pm 1.02$ |
| 32 | 20 | 63,964 | $20.64 \pm 0.05$ | $571.77 \pm 0.79$ | $7.64 \pm 0.24$ |
| 32 | 10 | 55,348 | $46.50 \pm 0.13$ | $778.80 \pm 1.12$ | $12.93 \pm 0.39$ |
| 32 | 4 | 49,948 | $121.55 \pm 0.35$ | $752.03 \pm 3.07$ | $13.99 \pm 0.41$ |
| 32 | 2 | 47,924 | $162.10 \pm 0.44$ | $767.17 \pm 2.38$ | $4.55 \pm 0.13$ |

Table 7: MSE loss (± the standard error of the mean) of OpenFOAM computations on test dataset of incompressible flow.

| Solver for $\boldsymbol{u}$ | Solver for $p$ | # division per unit length | $\Delta t$ | Total MSE ($\times 10^{-3}$) | Total time [s] |
|---|---|---|---|---|---|
| GAMG | Smooth | 22.5 | 0.050 | Divergent | Divergent |
| GAMG | Smooth | 22.5 | 0.010 | $6.09 \pm 0.02$ | $6.08 \pm 0.17$ |
| GAMG | Smooth | 22.5 | 0.005 | $6.04 \pm 0.02$ | $11.57 \pm 0.32$ |
| GAMG | Smooth | 22.5 | 0.001 | $4.80 \pm 0.02$ | $51.43 \pm 1.39$ |
| GAMG | Smooth | 45.0 | 0.050 | Divergent | Divergent |
| GAMG | Smooth | 45.0 | 0.010 | $0.46 \pm 0.00$ | $25.12 \pm 0.81$ |
| GAMG | Smooth | 45.0 | 0.005 | $0.78 \pm 0.00$ | $46.71 \pm 1.53$ |
| GAMG | Smooth | 45.0 | 0.001 | $1.04 \pm 0.00$ | $201.11 \pm 6.29$ |
| GAMG | Smooth | 90.0 | 0.050 | Divergent | Divergent |
| GAMG | Smooth | 90.0 | 0.010 | Divergent | Divergent |
| GAMG | Smooth | 90.0 | 0.005 | $0.15 \pm 0.00$ | $231.18 \pm 10.38$ |
| GAMG | GAMG | 22.5 | 0.050 | Divergent | Divergent |
| GAMG | GAMG | 22.5 | 0.010 | $6.05 \pm 0.02$ | $6.41 \pm 0.18$ |
| GAMG | GAMG | 22.5 | 0.005 | $6.00 \pm 0.02$ | $12.21 \pm 0.34$ |
| GAMG | GAMG | 22.5 | 0.001 | $4.80 \pm 0.02$ | $55.51 \pm 1.52$ |
| GAMG | GAMG | 45.0 | 0.050 | Divergent | Divergent |
| GAMG | GAMG | 45.0 | 0.010 | $0.46 \pm 0.00$ | $26.00 \pm 0.85$ |
| GAMG | GAMG | 45.0 | 0.005 | $0.77 \pm 0.00$ | $48.78 \pm 1.57$ |
| GAMG | GAMG | 45.0 | 0.001 | $1.03 \pm 0.00$ | $214.29 \pm 6.62$ |
| GAMG | GAMG | 90.0 | 0.050 | Divergent | Divergent |
| GAMG | GAMG | 90.0 | 0.010 | Divergent | Divergent |
| GAMG | GAMG | 90.0 | 0.005 | $0.14 \pm 0.00$ | $238.94 \pm 10.70$ |
| Smooth | Smooth | 22.5 | 0.050 | Divergent | Divergent |
| Smooth | Smooth | 22.5 | 0.010 | $5.59 \pm 0.02$ | $85.50 \pm 3.05$ |
| Smooth | Smooth | 22.5 | 0.005 | $5.41 \pm 0.02$ | $164.36 \pm 7.57$ |
| Smooth | Smooth | 22.5 | 0.001 | $4.19 \pm 0.02$ | $765.50 \pm 29.65$ |
| Smooth | Smooth | 45.0 | 0.050 | Divergent | Divergent |
| Smooth | Smooth | 45.0 | 0.010 | $51.10 \pm 0.05$ | $426.07 \pm 22.51$ |
| Smooth | Smooth | 45.0 | 0.005 | $2.09 \pm 0.00$ | $824.71 \pm 39.90$ |
| Smooth | Smooth | 45.0 | 0.001 | $1.12 \pm 0.00$ | $3960.88 \pm 151.93$ |
| Smooth | Smooth | 90.0 | 0.050 | Divergent | Divergent |
| Smooth | Smooth | 90.0 | 0.010 | Divergent | Divergent |
| Smooth | Smooth | 90.0 | 0.005 | $4493.78 \pm 1.88$ | $3566.05 \pm 183.75$ |

Table 8: Ablation study on 2D incompressible flow dataset. The value represents MSE loss ($\pm$ standard error of the mean) on the test dataset. "Divergent" means the implicit solver does not converge and the loss gets extreme value ($\sim 10^{14}$). This presents the same results as Table 3.

| Method | $\boldsymbol{u}$ ($\times 10^{-4}$) | $p$ ($\times 10^{-3}$) | $\hat{\boldsymbol{u}}_{\text{Dirichlet}}$ ($\times 10^{-4}$) | $\hat{p}_{\text{Dirichlet}}$ ($\times 10^{-3}$) |
|---|---|---|---|---|
| (A) Without encoded boundary | Divergent | Divergent | Divergent | Divergent |
| (B) Without boundary condition in the neural nonlinear solver | $65.10 \pm 0.38$ | $21.70 \pm 0.09$ | $0.00 \pm 0.00$ | $0.00 \pm 0.00$ |
| (C) Without neural nonlinear solver | $31.03 \pm 0.19$ | $9.81 \pm 0.04$ | $\mathbf{0.00} \pm 0.00$ | $\mathbf{0.00} \pm 0.00$ |
| (D) Without boundary condition input | $20.08 \pm 0.21$ | $3.61 \pm 0.02$ | $59.60 \pm 0.89$ | $1.43 \pm 0.05$ |
| (E) Without Dirichlet layer | $8.22 \pm 0.07$ | $1.41 \pm 0.01$ | $18.20 \pm 0.28$ | $0.38 \pm 0.01$ |
| (F) Without pseudoinverse decoder | $8.91 \pm 0.06$ | $2.36 \pm 0.02$ | $1.97 \pm 0.06$ | $\mathbf{0.00} \pm 0.00$ |
| (G) Without pseudoinverse decoder with Dirichlet layer after decoding | $6.65 \pm 0.05$ | $1.71 \pm 0.01$ | $\mathbf{0.00} \pm 0.00$ | $\mathbf{0.00} \pm 0.00$ |
| **PENN** | $\mathbf{4.36} \pm 0.03$ | $\mathbf{1.17} \pm 0.01$ | $\mathbf{0.00} \pm 0.00$ | $\mathbf{0.00} \pm 0.00$ |

# D   Experiment details: advection-diffusion dataset

To test the generalization ability of PENNs regarding PDE's parameters and time series, we run an experiment with the advection-diffusion dataset. The governing equation regarding the temperature field $T$ used for the experiment is expressed as:

$$\frac{\partial T}{\partial t} = -c \begin{pmatrix} 1 \\ 0 \\ 0 \end{pmatrix} \cdot \nabla T + D\nabla \cdot \nabla T \qquad (t, \boldsymbol{x}) \in (0,1) \times \Omega, \qquad (63)$$

$$T(t=0, \boldsymbol{x}) = 0 \qquad \boldsymbol{x} \in \Omega, \qquad (64)$$

$$T = \hat{T} \qquad (t, \boldsymbol{x}) \in \partial\Omega_{\text{Dirichlet}}, \qquad (65)$$

$$\nabla T \cdot \boldsymbol{n} = 0 \qquad (t, \boldsymbol{x}) \in \partial\Omega_{\text{Neumann}}, \qquad (66)$$

where $c \in \mathbb{R}$ is the magnitude of a known velocity field, and $D \in \mathbb{R}$ is the diffusion coefficient. We set $\Omega = \{\boldsymbol{x} \in \mathbb{R}^3 \mid 0 < x_1 < 1 \wedge 0 < x_2 < 1 \wedge 0 < x_3 < 0.01\}$, $\partial\Omega_{\text{Dirichlet}} = \{\boldsymbol{x} \in \partial\Omega \mid x_1 = 0\}$ and $\partial\Omega_{\text{Neumann}} = \partial\Omega \setminus \partial\Omega_{\text{Dirichlet}}$.

## D.1   Dataset

We varied $c$ and $D$ from 0.0 to 1.0, eliminating the condition $c = D = 0.0$ because nothing drives the phenomena, and and varied $\hat{T}$ from 0.1 to 1.0. Like the incompressible flow dataset, we generated fine meshes, ran computation with OpenFOAM, and interpolated the obtained temperature fields onto coarser meshes. We split the generated data into training, validation, and test dataset containing 960, 120, and 120 samples. The dataset is uploaded online.[8]

## D.2   Model architecture

The strategy to construct PENN for the advection-diffusion dataset is consistent with one for the incompressible flow dataset (see Appendix C.3). The input features of the model are:

- $T(t = 0.0)$: The initial temperature field
- $\hat{T}$: The Dirichlet boundary condition for the temperature field
- $(c, 0, 0)^\top$: The velocity field
- $c$: The magnitude of the velocity
- $D$: The diffusion coefficient
- $e^{-0.5d}, e^{-1.0d}, e^{-2.0d}$: Features computed from $d$, the distance from the Dirichlet boundary

and the output features are:

- $T(t = 0.25)$: The temperature field at $t = 0.25$
- $T(t = 0.50)$: The temperature field at $t = 0.50$
- $T(t = 0.75)$: The temperature field at $t = 0.75$
- $T(t = 1.00)$: The temperature field at $t = 1.00$

The encoded governing equation is expressed as:

$$h_T(t + \Delta t, \boldsymbol{x}) = h_T(t, \boldsymbol{x}) + \mathcal{D}_{\text{NIsoGCN;A-D}}(h_T)(t + \Delta t, \boldsymbol{x}) \qquad (67)$$

$$\mathcal{D}_{\text{NIsoGCN;A-D}}(h_T) := -\boldsymbol{h_c} \cdot \text{NIsoGCN}_{0\rightarrow 1}(h_T) + h_D \, \text{NIsoGCN}_{0\rightarrow 1\rightarrow 0}(h_T) \qquad (68)$$

The corresponding neural nonlinear solver is:

$$h_T^{(i+1)} = h_T^{(i)} - \alpha_{\text{BB}}^{(i)} \left[ h_T^{(i)} - h_T^{(0)} - \mathcal{D}_{\text{NIsoGCN;A-D}}(h_T^{(i)})\Delta t \right], \qquad (69)$$

Because the task is to predict time series data, we adopt autoregressive architecture for the nonlinear neural solver, i.e., input the output of the solver of the previous step (which is in the encoded space) to predict the encoded feature of the next step (see Figure 13). Figures 14 and 15 present the detailed architecture of the PENN model for the advection-diffusion dataset experiment.

To confirm the PENN's effectiveness, we ran the ablation study similar to that in the incompressible flow dataset. The training is performed for up to ten hours using the Adam optimizer for each setting.

---

[8] https://savanna.ritc.jp/~horiem/penn_neurips2022/data/ad/ad_preprocessed.tar.gz

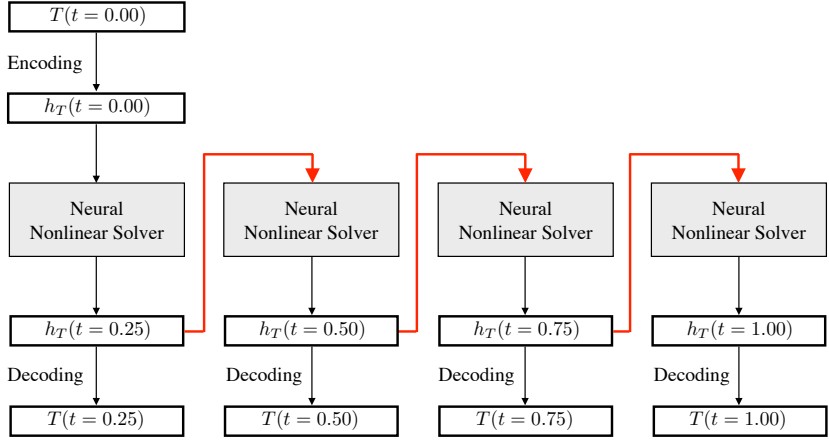

Figure 13: The concept of the neural nonlinear solver for time series data with autoregressive architecture. The solver's output is fed to the same solver to obtain the state at the next time step (bold red arrow). Please note that this architecture can be applied to arbitrary time series lengths.

Table 9: MSE loss ($\pm$ the standard error of the mean) on test dataset of the advection-diffusion dataset.

| Method | $T$ ($\times 10^{-4}$) | $\hat{T}_{\mathrm{Dirichlet}}$ ($\times 10^{-4}$) |
|---|---|---|
| (A) Without encoded boundary | $54.191 \pm 6.36$ | $0.0000 \pm 0.0000$ |
| (B) Without boundary condition in the neural nonlinear solver | $390.828 \pm 24.58$ | $0.0000 \pm 0.0000$ |
| (C) Without neural nonlinear solver | $6.630 \pm 1.21$ | $0.0000 \pm 0.0000$ |
| (D) Without boundary condition input | $465.492 \pm 26.47$ | $868.7009 \pm 15.5447$ |
| (E) Without Dirichlet layer | $2.860 \pm 2.46$ | $1.1703 \pm 0.0328$ |
| (F) Without pseudoinverse decoder | $44.947 \pm 6.00$ | $9.7130 \pm 0.1201$ |
| (G) Without pseudoinverse decoder with Dirichlet layer after decoding | $4.907 \pm 4.87$ | $0.0000 \pm 0.0000$ |
| **PENN** | $\mathbf{1.795} \pm 1.33$ | $0.0000 \pm 0.0000$ |

## D.3 Results

Table 9 presents the results of the ablation study. As well as the incompressible flow dataset, we found that the PENN model with all the proposed components achieved the best performance. Because the boundary condition applied is relatively simple compared to the incompressible flow dataset, the configuration without the Dirichlet layer (Model (E)) showed the second best performance; however, the fulfillment of the Dirichlet condition of that model is not rigorous.

Figures 16, 17, and 18 show the visual comparison of the prediction with the PENN model against the ground truth. As seen in the figures, one can see that our model is capable of predicting time series under various boundary conditions and PDE parameters, e.g., pure advection (Figure 16), pure diffusion (Figure 17), and mixed advection and diffusion (Figure 18).

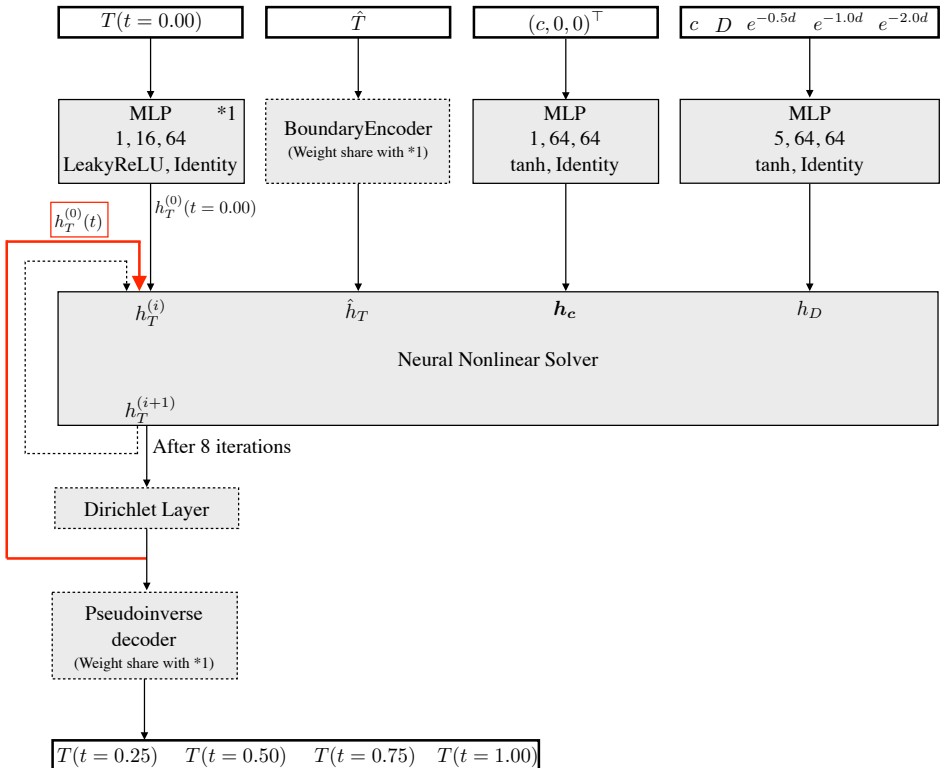

Figure 14: The overview of the PENN architecture for the advection-diffusion dataset. Gray boxes with continuous (dotted) lines are trainable (untrainable) components. Arrows with dotted lines correspond to the loop. In each cell, we put the number of units in each layer along with the activation functions used. The bold red arrow corresponds to the one in Figure 13.

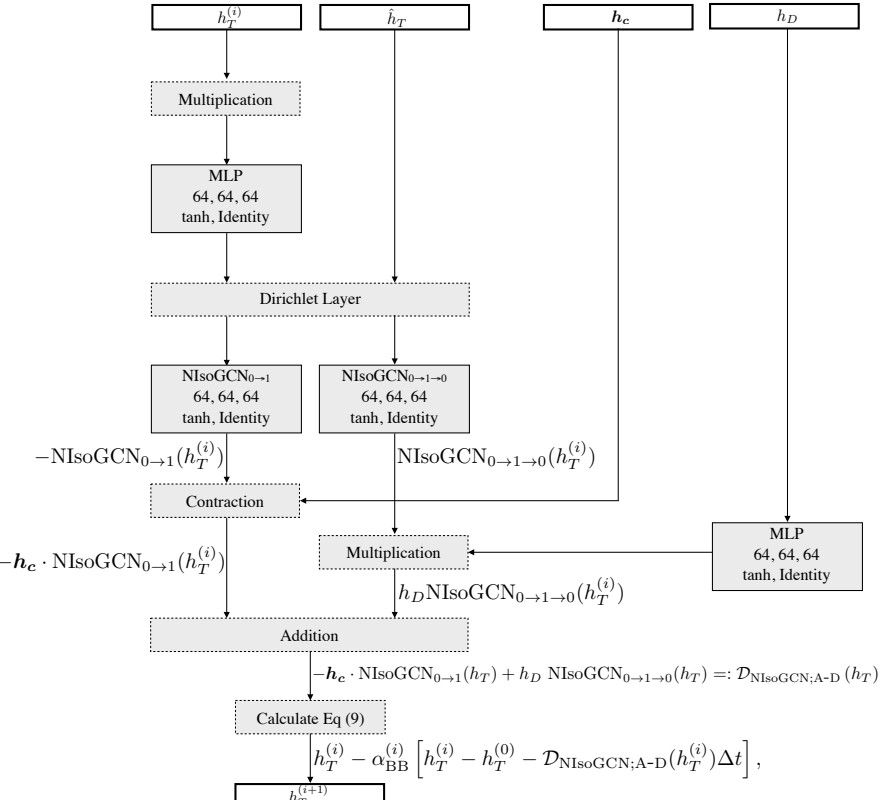

Figure 15: The overview of the PENN architecture for the advection-diffusion dataset. Gray boxes with continuous (dotted) lines are trainable (untrainable) components. In each cell, we put the number of units in each layer along with the activation functions used.

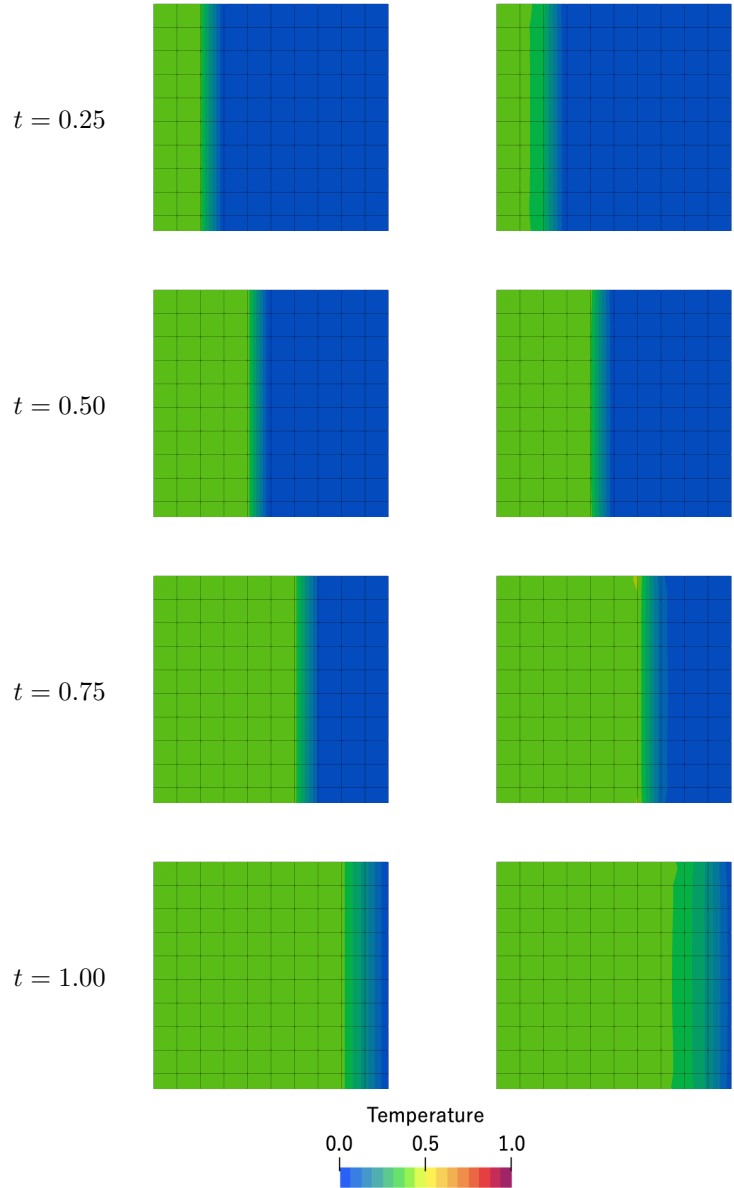

Figure 16: Visual comparison on a test sample between (left) ground truth obtained from OpenFOAM computation with fine spatial-temporal resolution and (right) prediction by PENN. Here, $c = 0.9$, $D = 0.0$, and $\hat{T} = 0.4$.

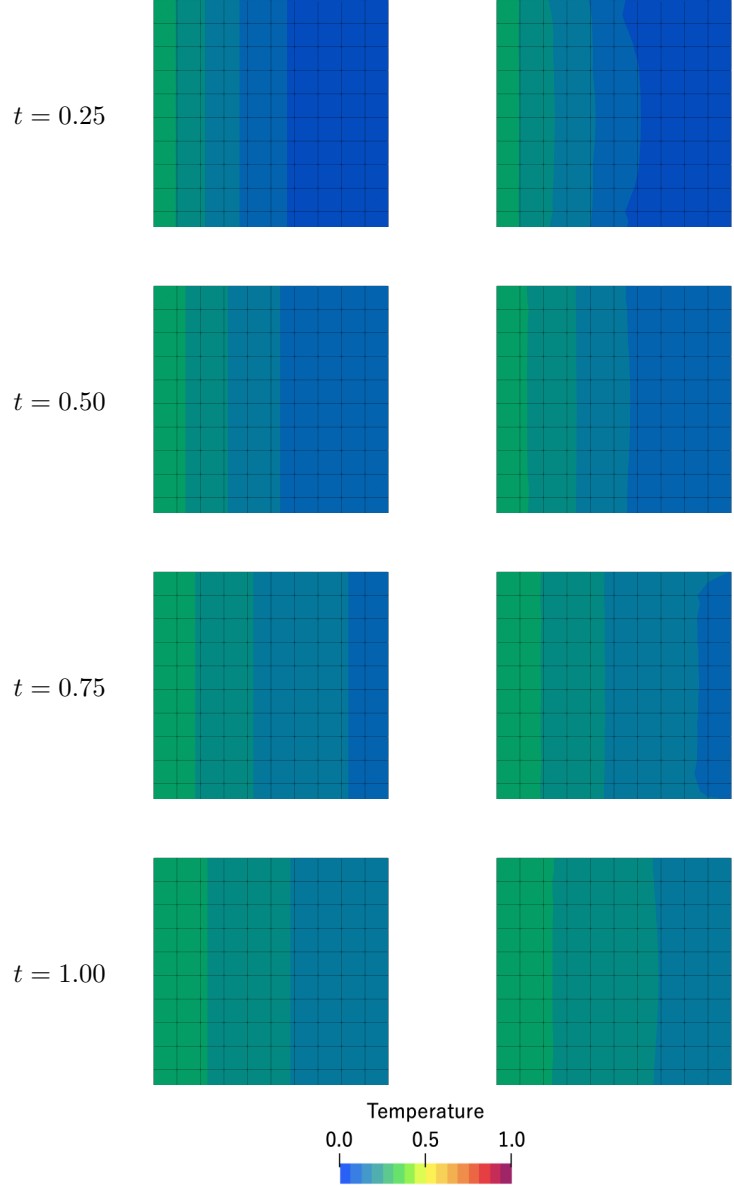

Figure 17: Visual comparison on a test sample between (left) ground truth obtained from OpenFOAM computation with fine spatial-temporal resolution and (right) prediction by PENN. Here, $c = 0.0$, $D = 0.4$, and $\hat{T} = 0.3$.

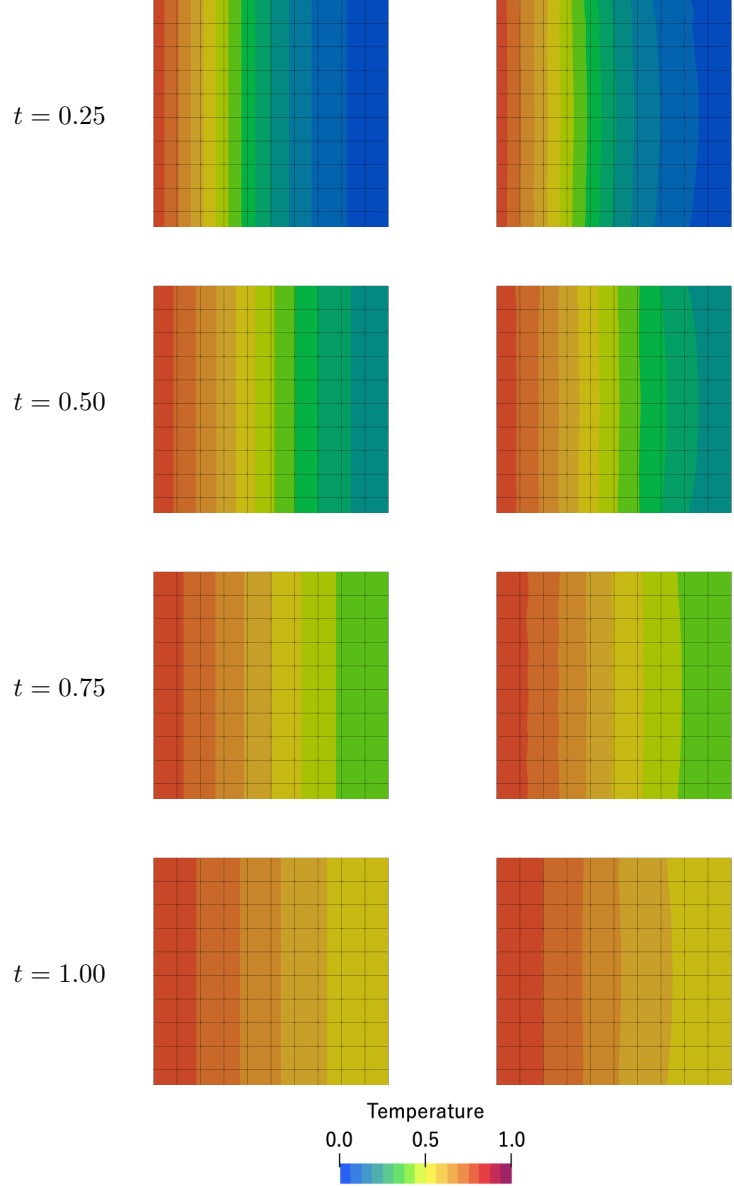

Figure 18: Visual comparison on a test sample between (left) ground truth obtained from OpenFOAM computation with fine spatial-temporal resolution and (right) prediction by PENN. Here, $c = 0.6$, $D = 0.3$, and $\hat{T} = 0.8$.