# OpenReview forum: "Physics-Embedded Neural Networks: Graph Neural PDE Solvers with Mixed Boundary Conditions"
_NeurIPS.cc/2022/Conference — NeurIPS 2022 Accept_

### Official Review · Reviewer_qCje · 2022-07-10

**Rating:** 7
**Confidence:** 3
**Soundness:** 3 good
**Presentation:** 4 excellent
**Contribution:** 3 good

**Summary:**

The authors proposed a novel design, building on top of IcoGCN to add two types of boundary conditions to the solution of the solver, Dirichlet conditions constraining the values of the functions at the boundaries, and Neumann conditions, constraining the gradients of the values at the boundaries. By leveraging the E(n) equivariant properties of the IcoGCN backbone design, the authors demonstrated improved performance compared to SOTA baselines such as MP-PDE, in particular with respect to translation and rotation of the domain. Compared to the original IcoGCN, the authors demonstrated that the proposed method is able to strictly satisfy Dirichlet and Neumann conditions, leading to improved accuracies of the solutions.

**Questions:**

One main thing I am trying to understand more deeply is how the proposed algorithm combines the first-principle physics with data priors. One way is to predict the results only using data prior, perhaps adding physics equations as an auxiliary loss (e.g., PINNs), which requires lots of training data, but leads to poor generalization. Another extreme is to only solve from the first principles, given as PDEs, using a PDE solver (as is the case in OpenFOAM), which requires no training data, but has very good generalization. In this work, the model does seem to require training on a given dataset (albeit a small dataset of only 203 examples), yet solves for a PDE using the Neural nonlinear solver in Sec 3.3. Please help me understand how the learned data prior is incorporated in this process?

Nit: Eqn (19) in the appendix has an extra parenthesis.

**Limitations:**

For the experimental evaluations, it would be interesting to see how the model generalizes to a different range of physical parameters, such as Reynolds number. Such evaluations seems to be lacking in the experiments.

**Strengths And Weaknesses:**

Strengths:
* Embedding the physical constraints into the design of the model itself by combining the PDE solution process with the model is an elegant and more generalizable approach to enforcing physical constraints, compared to simply using a loss penalty as an auxiliary loss only during training time.
* By incorporating the IcoGCN backbone, the authors are able to demonstrate translational and rotational equivariance of the final model, which is a very strong and desirable property missing in many works in the physics informed machine learning literature.
* The experimental evaluations are compelling. Not only did the authors compare with two reasonable baselines to demonstrate improved enforcement of boundary conditions and rotation/translation invariance, they also performed a runtime analysis between speed and accuracy tradeoffs, even considering the OpenFoam as a baseline. This is not too commonly evaluated among physics informed ML literature and I am happy to see it included in this work.
* The paper is clear and well written. In particular, I enjoyed the writeup for the backgrounds section, which doesn't assume too much prior knowledge on the subject matter and is easy to follow.

Weaknesses:
* Though the authors branded the novelty of this work on the E(n)-Equivariant properties of the model, it is not a unique contribution from this work since it is based off of the IcoGCN work which this work used as a backbone. Stressing this property (even in the title) feels like an oversell to me.
* Though the overall accuracy / runtime tradeoff for the proposed model is compelling (Fig. 4), I would like to see some thoughtful redesign of the model to allow flexibly adjusting accuracy / runtime also for this learned model. One suggestion is to use the results from this model to initialize a coupled PDE solver, so that with a good guess leveraging data prior, the model can achieve the same guaranteed accuracy at a faster speed.

---

> ### Author Response · Authors · 2022-08-02
> **Response to Reviewer qCje**
>
> We thank the reviewer for the constructive feedback and thoughtful questions.
>
> > Though the authors branded the novelty of this work on the E(n)-Equivariant properties of the model, it is not a unique contribution from this work since it is based off of the IcoGCN work which this work used as a backbone. Stressing this property (even in the title) feels like an oversell to me.
>
> The title is fixed to eliminate $\mathrm{E}(n)$-equivariance as advised. However, it is noteworthy that all the components we added (e.g., NeumannIsoGCN, the neural nonlinear solver) are fully compatible with $\mathrm{E}(n)$-equivariance. It is easy to break equivariance because if a portion of the model is not equivariant, the entire model will not be equivariant.
>
> > Though the overall accuracy / runtime tradeoff for the proposed model is compelling (Fig. 4), I would like to see some thoughtful redesign of the model to allow flexibly adjusting accuracy / runtime also for this learned model. One suggestion is to use the results from this model to initialize a coupled PDE solver, so that with a good guess leveraging data prior, the model can achieve the same guaranteed accuracy at a faster speed.
>
> We have added the results of the PENN models with varying numbers of parameters and iterations in the neural nonlinear solver, which adjusts the speed-accuracy tradeoff (Figure 4 and Table 5).
>
> The suggestion made by the reviewer seems quite attractive, as it can guarantee accuracy at the same level as the classical solvers. However, it is beyond the scope of this study, as our primary purpose was to construct an end-to-end neural PDE solver.
>
> > In this work, the model does seem to require training on a given dataset (albeit a small dataset of only 203 examples), yet solves for a PDE using the Neural nonlinear solver in Sec 3.3. Please help me understand how the learned data prior is incorporated in this process?
>
> Two main parts incorporate the first-principles physics into our machine learning model. Generally, we guarantee physics using inductive biases in the model architecture and utilize data to accelerate the prediction. Here are the details:
>
> * Model architecture: As discussed in Appendix C.3, which we updated with more details, we construct the PENN model to reflect the encoded governing equation. This works as a good inductive bias in the model, as it respects how physical quantities interact with each other and react with respect to coordinate transformation (i.e., the tensor transformation rule).
> * Training: As the reviewer pointed out, classical solvers do not require training data while keeping the prediction physical. However, classical solvers occasionally take a long time to compute because they always try to predict zero-based, i.e., utilizing no information. In extreme cases (even when we run the same analysis twice) a classical solver does not perform faster the second time. Our approach utilizes data for fast prediction while maintaining physics as much as possible. Thus, we encode the input physical quantities to higher dimensional space to utilize the neural nets' capability. During training, the model learns an effective way to encode and compute the PDEs in the encoded space.
>
> > Nit: Eqn (19) in the appendix has an extra parenthesis.
>
> Thank you for pointing this out. We fixed the equation.
>
> > For the experimental evaluations, it would be interesting to see how the model generalizes to a different range of physical parameters, such as Reynolds number. Such evaluations seems to be lacking in the experiments.
>
> We have added another experiment on the advection-diffusion dataset (Appendix D). In this experiment, we varied the velocity magnitude and diffusion coefficient. The results demonstrated that the model can learn and predict phenomena using various parameters. However, the parameters in the test dataset are within the domain of those in the training dataset.
>
> Generalizing the model to a parameter range outside the training dataset remains an open question because the learned data may not help predict a solution for such a parameter domain, and this would be the next direction of the research. However, it is beyond the scope of this paper, as we do not claim generalization for unseen parameters. However, the proposed model is still helpful because it can predict various states in various shapes, as it successfully predicts the test dataset for fluid phenomena. These results are supported by the novelties of the present work, which are the reliable treatment of boundary conditions and the neural nonlinear solver.

---

### Official Review · Reviewer_tb7f · 2022-07-10

**Rating:** 5
**Confidence:** 3
**Soundness:** 3 good
**Presentation:** 2 fair
**Contribution:** 3 good

**Summary:**

The authors’ primary contribution is the development of neural network layers that enable the treatment of Dirichlet- and Neumann-type boundary conditions in encoded space (the “boundary encoder”, “Dirichlet layer”, “NeumannIsoGCN layer” and the “pseudoinverse decoder”). For Dirichlet-type BCs, they accomplish this by constructing a decoder design such that the entire model will learn to approximate the identity function at the Dirichlet portion of the boundary (the “Dirichlet layer” and “pseudoinverse decoder”). The authors deal with Neumann-type BCs by extending the IsoGCN layer from the prior art to include terms that convert the Neumann BC into a penalty/constraint embedded into the architecture of the layer (“NeumannIsoGCN”).

A secondary contribution of the paper is the augmentation of a neural nonlinear solver where, by computing the “optimal step size” for a gradient descent step, they achieve a filter analogous to global pooling. This aids the neural model in learning highly nonlinear dynamics, such as exhibited by the incompressible Navier-Stokes equations.

These constructions are supported both by experiments and comparisons with the prior art as well as an ablation study to show the merit of each proposed component.


**Questions:**

1) The discussion of PINNs in section 2.2.1 strikes me as unrelated to the current work, or at least the connection isn’t very clear to me. I would advise the authors to clarify the limitation of PINNs they are discussing and make the connection or contrast with the current work explicit.

2) The same, but to a lesser degree, for GNNs in section 2.2.2. Please clarify the difference to the current work.

3) It seems to me that the main paper omits rather important details that are given in the first paragraph of section C.3 (lines 461-469 in the supplementary PDF). I realize that the page limit makes it difficult to include all the information, but consider finding a place for some or all of this information in the main paper.

4) How does the proposed method perform on domains that were not seen in the training set?


**Ethics Review Area:**

["I don’t know"]

**Limitations:**

No limitations.

**Strengths And Weaknesses:**

Strengths

* The paper is generally sound and the proposed model indeed outperforms the prior art model under comparison.

* The model achieves zero error on the Dirichlet boundary, as claimed.

* The ablation study can serve to show that each proposed component indeed contributes to the performance of the PENN model.

* Also of note is Figure 1 which presents an overview of the main contributions in a clean, legible and aesthetically pleasing way.

Weaknesses

* The title of the work appears to be slightly misleading in the sense that E(n)-equivariance stems from the work of Horie et al (2021). and not from the current work. I would suggest choosing a more descriptive title, such as “Physics-Embedded Neural Networks for Nonlinear Dynamics with Mixed Boundary Conditions”.

* I cannot find any information on the actual network architecture that is used. How is the domain encoded into the network? How flexible can the domain be? How about supporting heterogeneous coefficients?  It seems like the authors refer the reader to previous works, but I would expect this info to be present at the current paper or at least in the supplementary material. Also, from a brief look I couldn't find the architecture in the authors' code (but maybe I'm wrong). In any case - I do not see how one can replicate the results in this work.

* I would expect more comparisons to other NNs and classical solvers, particularly FEM/FVM and AMG-based solvers. The authors claim that they have tested their model against classical solvers, but I found no evidence of this in the main paper or supplementary PDF.

* I find it difficult to trust comparisons that do not detail metrics such as a number of trainable parameters or FLOPs. As it currently stands, I am unable to judge the tradeoff I would be making by using PENN as opposed to MP-PDE, for example. Consider adding these metrics as their absence makes true comparison very difficult. Figure 4 only partially addresses this concern but it does not present any speed/accuracy tradeoff for PENN (like it does for OpenFOAM and MP-PDE) and any speed/accuracy/parameter-count tradeoff is not considered at all.

---

> ### Author Response · Authors · 2022-08-02
> **Response to Reviewer tb7f (part 1)**
>
> We thank the reviewer for the constructive feedback and thoughtful questions.
>
> > The title of the work appears to be slightly misleading in the sense that E(n)-equivariance stems from the work of Horie et al (2021). and not from the current work. I would suggest choosing a more descriptive title, such as "Physics-Embedded Neural Networks for Nonlinear Dynamics with Mixed Boundary Conditions".
>
> Thank you for this suggestion. As advised, we omitted equivariance from the title and included the words "Mixed Boundary Conditions." We kept the phrase "Graph Neural PDE Solvers" because we think the word "PDE" is commonly used and understood; thus, using "PDE: may be helpful for readers to search for a good PDE solver.
>
> As pointed out, the main part of the equivariance comes from the work of Horie et al. (2021). Nevertheless, one of our contributions is to demonstrate that all the components we added (e.g., NeumannIsoGCN, neural nonlinear solver) are fully compatible with $\mathrm{E}(n)$-equivariance. Furthermore, it is easy to break equivariance as if a portion of the model is not equivariant, the entire model will not be equivariant.
>
> > I cannot find any information on the actual network architecture that is used. How is the domain encoded into the network? How flexible can the domain be? How about supporting heterogeneous coefficients? It seems like the authors refer the reader to previous works, but I would expect this info to be present at the current paper or at least in the supplementary material.
>
> We have added detailed explanations and figures of the actual network architectures (Figures 5, 9, 10, 11, 13, 14, and 15). We constructed the model using components that accept arbitrary input lengths (e.g., pointwise MLPs, deep sets, NeumannIsoGCNs) (Appendix C.4, lines 544-546). Therefore, our model is flexible in accepting arbitrary meshes as the inputs. Although we have not demonstrated heterogeneous coefficients in this study, the model can deal with this condition by feeding the heterogeneous features as inputs, as we used $e^{-0.5d}$ in the incompressible flow (Appendix C.3, line 513).
>
> >  Also, from a brief look I couldn't find the architecture in the authors' code (but maybe I'm wrong). In any case - I do not see how one can replicate the results in this work.
>
> The model architectures are stored in the YAML files in the following directories in the supplementary material:
>
> * penn_neurips2022_supplemental_20220803/data/grad: The gradient dataset
> * penn_neurips2022_supplemental_20220803/data/fluid: The incompressible flow dataset
> * penn_neurips2022_supplemental_20220803/data/ad directories: The advection-diffusion dataset (added in the latest update)
>
> We see that it was unclear to readers; thus, we have added some details  where the model architecture is described to README.md.
>
> > I would expect more comparisons to other NNs and classical solvers, particularly FEM/FVM and AMG-based solvers. The authors claim that they have tested their model against classical solvers, but I found no evidence of this in the main paper or supplementary PDF.
>
> We believe that the MP-PDE is the best NN model to compare to, owing to the following reasons:
>
> * It can deal with various boundary conditions in a general manner.
> * It can deal with various shapes leveraging GNN's feature.
> * It is contemporary and sufficiently powerful, as it is published in ICLR 2022.
>
> To the best of our knowledge, we were not able to find any other NN models that can satisfy all of the above reasons.
>
> We used OpenFOAM, which adopts FVM, with AMG-based solvers. In general, the FEM tends to exhibit instability (e.g., spurious oscillation) and takes time to solve fluid problems. In the literature, Molina-Aiz et al.[1] reported that the FEM requires twice as long computation time as compared to the FVM. Regarding the linear solver, we have added a comparison with the following configurations (Table 6):
>
> * AMG solver for $p$ and the smooth solver for $\boldsymbol{u}$ (our initial choice)
> * AMG solver for $p$ and $\boldsymbol{u}$
> * The smooth solver for $p$ and $\boldsymbol{u}$
>
> The results confirm that our initial choice (AMG for p and smooth for u) was optimal. Also, all the speed-accuracy data used to plot Figure 4 were added (Tables 5, 6, and 7).
>
> [1] Molina-Aiz, F. D., Hicham Fatnassi, Thierry Boulard, Jean-Claude Roy, and D. L. Valera. "Comparison of finite element and finite volume methods for simulation of natural ventilation in greenhouses." Computers and electronics in agriculture 72, no. 2 (2010): 69-86.

---

> > ### Author Response · Authors · 2022-08-02
> > **Response to Reviewer tb7f (part 2)**
> >
> > > I find it difficult to trust comparisons that do not detail metrics such as a number of trainable parameters or FLOPs. As it currently stands, I am unable to judge the tradeoff I would be making by using PENN as opposed to MP-PDE, for example. Consider adding these metrics as their absence makes true comparison very difficult.
> >
> > We have added tables presenting the speed, accuracy, and number of parameters (if any) as Tables 5, 6, and 7. Also, we have added a discussion related to the parameter count (lines 563-568). As seen in the table, the PENN models have a significantly smaller number of parameters than the MP-PDE models, with one to two digits without degrading the predictive performance. This is because our model effectively shares the parameters in the neural nonlinear solver, where the same network is used for each iteration.
> >
> > > Figure 4 only partially addresses this concern but it does not present any speed/accuracy tradeoff for PENN (like it does for OpenFOAM and MP-PDE) and any speed/accuracy/parameter-count tradeoff is not considered at all.
> >
> > We have added an additional study by varying the number of parameters for the PENN and MP-PDE models (Figure 4 and Tables 5, 6, and 7). Through this comparison, we demonstrated that our method improves the speed-accuracy tradeoff for multiple configurations. We also found that decreasing the number of iterations in the neural nonlinear solver significantly affected the computation time compared to the number of parameters.
> >
> > > The discussion of PINNs in section 2.2.1 strikes me as unrelated to the current work, or at least the connection isn't very clear to me. I would advise the authors to clarify the limitation of PINNs they are discussing and make the connection or contrast with the current work explicit.
> >
> > We have added a line to contrast our model against PINNs (lines 101 and 102). The advantage of our model is its ability to generalize shapes, translation, and rotation as it does not consider the absolute positions of the vertices as inputs. The limitations of PINNs are as follows (lines 94-100):
> >
> > * Generalization: PINNs need to take the absolute positions of the vertices to leverage automatic differentiation regarding space. If we input the absolute positions into NNs, to learn physical quantities as functions of space, generalization regarding shapes and boundary conditions will deteriorate because the learned functions may not work for problems with different shapes and boundary conditions.
> > * Guarantee of physics: PINNs utilize physics information during training. However, the prediction of PINNs has less justification for physics, because typical PINNs have no inductive bias, inside the model, to guarantee physics. Our model has physics embedded, thus generating more reliable predictions.
> >
> > > The same, but to a lesser degree, for GNNs in section 2.2.2. Please clarify the difference to the current work.
> >
> > We have added some sentences that distinguish our model from other GNNs (lines 112-114). The superiority of our method lies in the proposed neural nonlinear solver, which considers global interaction in an effective, efficient, and $\mathrm{E}(n)$-equivariant way. Furthermore, most GNNs use local connections with a fixed number of message passings and do not consider global interactions.
> >
> > > It seems to me that the main paper omits rather important details that are given in the first paragraph of section C.3 (lines 461-469 in the supplementary PDF). I realize that the page limit makes it difficult to include all the information, but consider finding a place for some or all of this information in the main paper.
> >
> > We moved most of the contents concerned to the main paper, and we hope that it is now easier to comprehend the overview of the actual machine learning models we used.

---

> > > ### Author Response · Authors · 2022-08-02
> > > **Response to Reviewer tb7f (part 3)**
> > >
> > > > How does the proposed method perform on domains that were not seen in the training set?
> > >
> > > Our model can accept any mesh as input, as discussed above, resulting in generalization with various analysis domains. In particular, the model has $\mathrm{E}(n)$-equivariance; thus, it can predict phenomena on unseen analysis domains with both translation and rotation.
> > >
> > > Regarding the parameter domain (e.g., the Reynolds number), the focus should be directed to our new experiments on the advection-diffusion dataset (Appendix D). The model can predict test data with various parameters (velocity and diffusion coefficient). However, this was within the range of the parameters in the training dataset.
> > >
> > > Generalizing the model to a parameter range outside the training dataset remains an open question because the learned data may not help predict a solution for such a parameter domain, and this would be the next direction of the research. However, it is beyond the scope of this paper, as we do not claim generalization for unseen parameters. However, the proposed model is still helpful because it can predict various states in various shapes, as it successfully predicts the test dataset for fluid phenomena. These results are supported by the novelties of the present work, which are the reliable treatment of boundary conditions and the neural nonlinear solver.

---

### Official Review · Reviewer_ZHC4 · 2022-07-10

**Rating:** 7
**Confidence:** 4
**Soundness:** 3 good
**Presentation:** 3 good
**Contribution:** 3 good

**Summary:**

This work proposed to use E(n)-Equivariant Graph Neural Network to solve PDEs. Main differences compared to existing works include 1) using a boundary node encoder and pseudoinverse decoder to enforce boundary conditions and achieve better long-term accuracy. 2) embed the PDE inside the model to achieve better long-term accuracies, via the proposed NeumannIsoGCN

**Questions:**

- in eq) 17, does $i$ refer to time steps, or gradient descent steps? If it refers to # of time steps, does propose approach generalize dynamic simulation of arbitrary timesteps?

**Ethics Review Area:**

["I don’t know"]

**Limitations:**

My primary concern is also one point that I don't quite understand if the $i$ in eq. 17 refer to time steps. If so, how could it generalize to potentially longer time scales?

**Strengths And Weaknesses:**

***novelty*** The formulation is new and the global encoding and embedded PDE inside the GNN is novel, to the best of my knowledge.

 ***quality and clarity*** The work is well written and well-motivated with good background introductions and well-organized experimental results. The accuracy speed trade-off plot is interesting.

***significant*** The work demonstrates their work on solving an important task.

---

> ### Author Response · Authors · 2022-08-02
> **Response to Reviewer ZHC4**
>
> We thank the reviewer for the constructive feedback and thoughtful questions.
>
> > in eq) 17, does i refer to time steps, or gradient descent steps? If it refers to # of time steps, does propose approach generalize dynamic simulation of arbitrary timesteps?
>
> $i$ in Equation (17) refers to the step in the neural nonlinear solver (gradient descent steps), as we use an iterative method for the solver. We have clarified this in the manuscript (lines 192 and 193).
>
> > My primary concern is also one point that I don't quite understand if the i in eq. 17 refer to time steps. If so, how could it generalize to potentially longer time scales?
>
> As discussed above, $i$ is unrelated to the absolute time step. Therefore, our model can predict the future state regardless of the absolute time of the input state. A newly added experiment shows that our model can learn and predict time series data by applying the same neural nonlinear solver for each time step (Appendix D). Because of the autoregressive architecture of the model, it can generalize to time series data with arbitrary length.

---

> > ### Comment · Reviewer_ZHC4 · 2022-08-09
> > **Thanks**
> >
> > Thanks for the clarification.

---

### Official Review · Reviewer_AnrD · 2022-07-15

**Rating:** 6
**Confidence:** 2
**Soundness:** 3 good
**Presentation:** 2 fair
**Contribution:** 2 fair

**Summary:**

The paper presents a neural PDE solver based on adn encode-process-deode architecture that respects boundary conditions thanks to a novel GNN-based gradient operator. Other than the proposed version of an E(n)-equivariant GNN nonlinear solver they also propose a different encoding process for boundary condition treatments in the encoded space. Experiments comprise prediction of the gradient field from a given scalar field to verify the expressiveness of the proposed version of the GNN-based gradient operator; and the task of learning incompressible flow. Other than respecting boundary conditions by construction, results show important improvement with respect to the state of the art neural PDE solvers.

**Questions:**

Can you elaborate more on the Dirichlet encoder-decoder? I do not understand whether it is as simple as it looks or perhaps there is something more. My understanding is that the encoder distinguish betweenboundary nodes and nodes that are not on the boundary and just apply a the pseudoinverse transformation on boundary nodes for decoding. In this case I don’t understand its role in performing better predictions other than just trivially enforcing boundary conditions at the end of the processing.

**Limitations:**

Authors address potential societal impact of their work and mention the fact that the properties of the model limit its applicability domain. They mention that the proposed method is not suitable for solving inverse problems but don't elaborate much on that. The improvement with respect to the state of the art (here Brandstetter et al.) is significant. Perhaps a a few simple experiments similar to those in Brandstetter et al would give more information on whether improvement in predictions is really due to the constraints enforced by the proposed architecture.

**Strengths And Weaknesses:**

The idea seems to be very simple and based on existing methods but is effective. The connection to global pooling in computing the step size $\alpha$ of the Barzilai–Borwein method is also interesting and perhaps deserves some more discussions. The paper is sometimes hard to read. And not very clear.
e.g.,
3.2 Where it is explained that the weight should be kept small to 'respect' information in the neighborhood.
3.2 Doesn't explain how the model can be generalized to vectors or higher rank tensors. (edited)

Moreover, in the paper it doesn't really explain how experiments are performed so I assume the reader is often very familiar with this kind of experiments to test neural pde solvers otherwise they need toexplain it better. Especially how GNN gradient operators predict gradients. There are some more explanations in the Appendix but overall the description remains unclear.

---

> ### Author Response · Authors · 2022-08-02
> **Response to Reviewer AnrD (part 1)**
>
> We thank the reviewer for the constructive feedback and thoughtful questions.
>
> > The connection to global pooling in computing the step size α of the Barzilai–Borwein method is also interesting and perhaps deserves some more discussions.
>
> We have added a subsection on the Barzilai–Borwein method in Appendix A.3. In the method, $\alpha$ attempts to facilitate convergence as well as possible for every vertex. Thus, global information is included in $\alpha$.
>
> > The paper is sometimes hard to read. And not very clear. e.g., 3.2 Where it is explained that the weight should be kept small to 'respect' information in the neighborhood.
>
> We have expanded upon the explanation regarding the value of $w_i$ (lines 168-171). With an extremely large $w_i$, other terms tend to be neglected, relatively, leading to information in the neighborhood being disregarded.
>
> > 3.2 Doesn't explain how the model can be generalized to vectors or higher rank tensors.
>
> An explanation of how NeumannIsoGCN could be generalized to higher rank tensors is included in Appendix A.2, lines 441-445, where it can be generalized using a recursive definition as in Equation (29).
>
> > Moreover, in the paper it doesn't really explain how experiments are performed so I assume the reader is often very familiar with this kind of experiments to test neural pde solvers otherwise they need to explain it better.
>
> We have refined the descriptions of the experiments, particularly in Appendix C.3, and the input and output features used in the experiments were clarified. It is now clear that our model is in line with the typical formulation of neural PDE solvers described in the line 88, which takes the state at $t$ as inputs and the output state at $t + \Delta t$.
>
> > Especially how GNN gradient operators predict gradients.
>
> We have added an explanation of some connections between NeumannIsoGCN (NIsoGCN) and spatial differential operators in Appendix A.2, lines 446-456. These models are similar to the GCN, the model's origin, which involves multiplication between adjacency matrices, input features, and a trainable weight matrix. The output could be an encoded representation of the derivative if the model is well-trained.
>
> Also, we have added figures of the machine learning models. In particular, Figure 5 shows the architecture of the gradient dataset as that model only computes the gradient. In contrast, the models of the other experiments are somewhat complicated. For example, the NIsoGCN block, shown in Figure 5 (b), takes encoded features (scalar $\psi$ and the normal directional derivative $\hat{g}\boldsymbol{n}$) as inputs and outputs the gradient in the encoded space. These encoded gradients have 16 vectors for each vertex. The MLP block next to the NIsoGCN block decodes the 16 vectors to 1 vector per vertex to obtain the final prediction of the gradient.
>
> > Can you elaborate more on the Dirichlet encoder-decoder? I do not understand whether it is as simple as it looks or perhaps there is something more. My understanding is that the encoder distinguish between boundary nodes and nodes that are not on the boundary and just apply a the pseudoinverse transformation on boundary nodes for decoding.
>
> We have added Figure 9, which shows an overview of the model we used for the incompressible flow dataset. Each input feature was encoded separately; however, the networks are shared by the DirichletLayers. Once the nonlinear solver has completed, we apply the Dirichlet layers as in Equation (17), and then decode using the pseudoinverse decoders, which are applied to all the vertices in the mesh, not only on the boundary. Please note that the encoders and decoders are applied pointwise, as is done in the standard encode-process-decode architecture. We have added a description of this to Appendix C.4.
>
> >  In this case I don't understand its role in performing better predictions other than just trivially enforcing boundary conditions at the end of the processing.
>
> Because the pseudoinverse decoder is applied to all vertices, it facilitates the spatial continuity of the output, as mentioned in Appendix C.7 (lines 605-608). In addition, if there is no Dirichlet layer in the neural nonlinear solver loop, the hidden features tend to shift from what is expected, that is, the state satisfying the boundary condition. We added another model (model (B)) in the ablation study, which had no boundary condition assignment in the nonlinear solver and only had an assignment at the end of the network. Its performance was significantly worse than that of the proposed model (Tables 3 and 9), which corroborates our discussion.

---

> > ### Author Response · Authors · 2022-08-02
> > **Response to Reviewer AnrD (part 2)**
> >
> > > They mention that the proposed method is not suitable for solving inverse problems but don't elaborate much on that.
> >
> > We have added the reason why our model has difficulty solving inverse problems to Section 5, line 286. It states that our model uses the information of the available PDE, making our approach reliable and efficient. However, a typical inverse problem does not have an explicit form of PDE, thus making it difficult to utilize our model.
> >
> > > Perhaps a a few simple experiments similar to those in Brandstetter et al would give more information on whether improvement in predictions is really due to the constraints enforced by the proposed architecture.
> >
> > We have added a simple experiment using the advection-diffusion equation. The experiment on that dataset also showed that our proposed approach is more effective than the other ablation models. In addition, the results show that the PENN model can learn and predict time series data with various PDE parameters (flow velocity and diffusion coefficient).

---

### Author Response · Authors · 2022-08-02
**General comment**

We appreciate the feedback given by the reviewers. We have performed additional experiments and updated the manuscript. Here we summarize our main updates:

* We changed the title to "Physics-Embedded Neural Networks: Graph Neural PDE Solvers with Mixed Boundary Conditions" (although it seems that the tile on the OpenReview cannot be changed for the moment).
* We added detailed explanations regarding the machine learning model we constructed (Section 4.2.2, Appendix C.3)
* We performed additional parameter studies for PENN, MP-PDE, and OpenFOAM for a more comprehensive insight into the speed-accuracy tradeoff (Figure 4, Appendix C.6).
* We performed additional experiments using the advection-diffusion problems to demonstrate the capacity of the proposed model to predict time series data with various PDE parameters. (Appendix D)

We also note the minor updates we made:

* The appendix is included in the PDF file of the main paper.
* We added trainable weight in the definition of IsoGCN (Equation (10)) and NIsoGCN (Equation (14)), which was unintendedly missing in the first version.
* We switched to using the abbreviated name "NIsoGCN" for "NeumannIsoGCN" to save space.

We hope that our revised manuscript and responses could address all the questions and uncertainties.

---

### Meta-Review · Area_Chair_M9zb · 2022-08-26

**Recommendation:** Accept
**Confidence:** Certain

**Metareview:**

The paper proposes a E(n)-equivariant neural PDE solvers that can satisfy boundary conditions provably. The reviewers acknowledged the importance of the studied problem setting and generally appreciated the results. The paper is nicely written and provides both strong experimental results and theory. Indeed, a range of interesting experiments demonstrate the effectiveness of the proposed method. I want to thank the authors for their detailed responses that helped in answering some of the reviewers' questions. (The reviewers have provided detailed feedback in their reviews, and we strongly encourage the authors to incorporate this feedback when preparing a revised version of the paper.) In summary, this paper is a clear accept. Well done!

**Award:**

No

---

### Decision · Program_Chairs · 2022-09-14

Accept